# KNAPSACK RL: UNLOCKING EXPLORATION OF LLMS VIA OPTIMIZING BUDGET ALLOCATION

## ABSTRACT

Large Language Models (LLMs) can self-improve through reinforcement learning, where they generate trajectories to explore and discover better solutions. However, this exploration process is computationally expensive, often forcing current methods to assign limited exploration budgets to each task. This uniform allocation creates problematic edge cases: easy tasks consistently succeed while difficult tasks consistently fail, both producing zero gradients during training updates for the widely used Group Relative Policy Optimization (GRPO). We address this problem from the lens of exploration budget allocation. Viewing each task's exploration as an "item" with a distinct "value" and "cost", we establish a connection to the classical knapsack problem. From this, we derive an optimal assignment rule that transfers exploration budgets from easy tasks to challenging ones. When applied to GRPO, our method increases the effective ratio of non-zero policy gradients by 20–40% during training. As a computational "free lunch", it also enables substantially larger exploration budgets (e.g., 93 rollouts) for especially challenging tasks—budgets that would be computationally prohibitive under uniform allocation. These improvements translate to meaningful gains on mathematical reasoning benchmarks, with average improvements of 2–4 points and peak gains of 9 points on specific tasks. Notably, achieving comparable performance with traditional homogeneous allocation would require about 2x the computational resources.

## 1 INTRODUCTION

The remarkable capabilities of Large Language Models (LLMs) have led to their widespread application across various domains (OpenAI, 2025; Comanici et al., 2025; Anthropic, 2025; Meta, 2025; Yang et al., 2025). While pre-training on vast text corpora endows LLMs with general knowledge and linguistic fluency, fine-tuning them for specialized tasks often necessitates more targeted optimization beyond pre-training. Reinforcement Learning (RL) has emerged as a powerful paradigm for this purpose (Ouyang et al., 2022; Li et al., 2024; Guo et al., 2025), enabling LLMs to iteratively self-improve by interacting with environments. A popular instantiation is RL with verifiable rewards (Lambert et al., 2024), where LLMs generate responses and receive binary (true/false) feedback based on their outcomes, iteratively refining their internal policies to search for optimal solutions. Initially pioneered in mathematical reasoning (Jaech et al., 2024), this framework has since been extended to domains like coding (Luo et al., 2025a) and agentic tasks (Team et al., 2025).

A core challenge in these applications is *exploration*—sampling diverse trajectories to find better solutions. This process is computationally expensive in practice due to sequential nature of autoregressive generation. As such, most RL pipelines use a small number of rollouts per prompt (e.g., 8) for exploration. However, this uniform allocation strategy could lead to some problematic outcomes. For example, in the Group Relative Policy Optimization (GRPO) (Shao et al., 2024) algorithm, meaningful learning signals (gradients) only emerge when both successful and failed attempts are present in the same batch. With a uniform budget, easy tasks often result in all-success outcomes, and hard tasks in all-failure outcomes, leading to near-zero gradients and stalled learning. This issue has been well-documented in previous research (Yu et al., 2025; Chen et al., 2025a), and we approach it from the broader perspective of strategic exploration budget allocation.

We argue the fundamental problem is the mismatch between a task's difficulty and its assigned exploration budget. Hard tasks, which require extensive (could even require more than 100) exploration to find useful trajectories, receive too little effort under a uniform rule. Easy tasks, which require

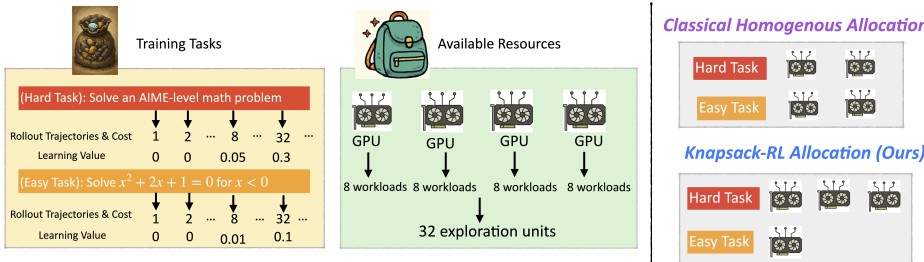

Figure 1: Illustration of our framework for allocating exploration budgets among tasks from computational resources. We model each task as an item with learning value and computational cost, then solve the allocation problem using Knapsack optimization.

minimal exploration, waste compute by being over-sampled. Thus, a heterogeneous and customized exploration allocation strategy is preferred.

To this end, we introduce a knapsack-based formulation: each task, when assigned a certain budget, can be conceptualized as an 'item' with an associated value (learning potential) and cost (computational effort of exploration). The allocation problem is thus equivalent to the classical knapsack problem (Mathews, 1896; Pisinger & Toth, 1998), where the objective is to maximize total value under a fixed global budget. We refer to this approach as Knapsack RL; see Figure 1 for illustration. When applied to the popular GRPO framework, our method enables a dynamic, heterogeneous allocation of exploration budgets, which allows sufficient exploration on training tasks.

Empirically, across Qwen series models (Yang et al., 2024; 2025) sized from 1B to 7B, we observe a 20-40% improvement in effective gradient ratios, translating into more reliable policy improvements and average performance gains of about 2-4 points on several challenging benchmarks. To get a better sense of this improvement, we note that achieving comparable improvements with uniform allocation would require nearly 2x the computation. We present this as a proof-of-concept, demonstrating a promising direction to boost the effectiveness of RL.

## 2 PRELIMINARY

Following (Ouyang et al., 2022; Shao et al., 2024), we model language generation as autoregressive sampling from a conditional probability distribution $\pi_\theta(y|x)$, where $x$ represents the input prompt and $y$ represents the generated response. The parameter $\theta$ denotes the trainable parameters. Our goal is to improve the language model via RL by maximizing the expected performance of responses generated from the model distribution $\pi_\theta$:

$$\max_\theta \mathbb{E}_{y \sim \pi_\theta(\cdot|x)}[r(x,y)] \tag{1}$$

In this paper, we focus on RL with verifiable rewards. Specifically, let $y = (\texttt{CoT}, \texttt{answer})$ denote the concatenation of Chain-of-Thought (CoT) (Wei et al., 2022) reasoning steps $\texttt{CoT}$ and the final solution $\texttt{answer}$. The reward function $r(x,y)$ is defined as:

$$r(x,y) = \mathbb{I}(\texttt{answer is correct with respect to } x), \tag{2}$$

where $\mathbb{I}(\cdot)$ is the indicator function and $r \in \{0,1\}$ is binary (1 for correct, 0 for incorrect). This outcome-based reward formulation has been widely adopted (see e.g., (Guo et al., 2025) and references therein) and has been shown to effectively incentivize reasoning abilities (Wen et al., 2025).

---

**Algorithm 1** RL with Classical Homogeneous Budget Allocation

---

1: **for** iteration $t = 1, 2, \ldots$ **do**
2:     Sample a mini-batch of prompts $(x_1, \ldots, x_M)$
3:     Generate $N$ responses for each prompt $x_i$           ▷ *Budget Allocation*
4:     Evaluate the rewards (e.g., Equation (2)) and compute the gradients (e.g., Equation (3))
5:     Update model parameters with estimated gradients

---

To optimize Equation (1), policy gradient methods (Sutton et al., 1999) are commonly employed. Among these, REINFORCE (Williams, 1992)-style stochastic policy gradient methods have become standard since (Li et al., 2024). These methods stochastically sample $N$ responses from $\pi_\theta$ and estimate gradients using direct reward feedback. Originally designed for single-task RL, this approach

is typically extended to multi-task RL by employing homogeneous exploration budget allocation. Algorithm 1 summarizes this classical framework.

In Algorithm 1, the sampling process in Line 3 corresponds to exploration in RL, where the model generates responses to search for optimal solutions. Line 5 corresponds to exploitation, updating the model to leverage feedback from data. We adopt the widely used gradient estimator from Group Relative Policy Optimization (GRPO) (Shao et al., 2024):

$$g(\theta) = \sum_{i=1}^{M} \sum_{j=1}^{N} \nabla_\theta \log \pi_\theta(y_{ij}|x_i) \cdot (r(x_i, y_{ij}) - b_i) \cdot c_i \tag{3}$$

where $y_{ij}$ denotes the $j$-th sampled response for prompt $x_i$, and $\nabla_\theta \log \pi_\theta(y_{ij}|x_i)$ represents the gradient of the log-probability with respect to model parameters $\theta$. The baseline $b_i$ and normalization factor $c_i$ are defined as: $b_i = 1/N \cdot \sum_{j=1}^{N} r(x_i, y_{ij})$ and $c_i = 1/(\sigma_i + \epsilon)$ with $\sigma_i = \sqrt{1/N \cdot \sum_{j=1}^{N}(r(x_i, y_{ij}) - b_i)^2}$ is the standard deviation of rewards for prompt $x_i$, and $\epsilon$ is a small constant ($10^{-6}$) preventing division by zero when $\sigma_i = 0$. Technically, GRPO computes relative advantages within each response group (prompt), increasing likelihood of positive responses and decreasing likelihood of negative ones.

## 3 DIAGNOSING EXPLORATION IN HOMOGENEOUS BUDGET ALLOCATION

In this section, we discuss the limitations of homogeneous budget allocation for GRPO and present empirical observations that motivate our work.

### 3.1 MOTIVATION

Exploration in RL is computationally expensive due to the sequential nature of autoregressive generation, often requiring substantial GPU memory and hours of computation, especially for reasoning tasks. Thus, it is critical to assess how much each collected sample actually contributes to gradient updates. For GRPO, we make the following observation.

**Observation 1.** *Let $g_i = \sum_{j=1}^{N} \nabla_\theta \log \pi_\theta(y_{ij}|x_i) \cdot (r(x_i, y_{ij}) - b_i) \cdot c_i$ be the gradient for prompt $i$. If $\sigma_i = 0$, meaning that all $N$ sampled responses for $x_i$ yield identical rewards (all correct or all incorrect), then $(r(x_i, y_{ij}) - b_i) = 0$ for every sample, leading to $g_i = 0$. In this case, the model receives no learning signal from that prompt.*

This phenomenon is widely recognized as a major bottleneck for GRPO in practice (Yu et al., 2025; Chen et al., 2025a). To formally track it, we introduce the metric `effective-gradient-ratio`, which measures the proportion of individual samples that contribute non-zero gradients:

$$\texttt{effective-gradient-ratio} = \frac{1}{M \cdot N} \sum_{i=1}^{M} \sum_{j=1}^{N} \mathbb{I}(g_{i,j} \neq 0), \tag{4}$$

where $g_{i,j} = \nabla_\theta \log \pi_\theta(y_{ij}|x_i) \cdot (r(x_i, y_{ij}) - b_i) \cdot c_i$ is the gradient contribution from the $j$-th sample of the $i$-th prompt. A higher value indicates that a larger fraction of samples are contributing useful learning signals. We also define two complementary metrics: `zero-gradient-ratio (by all positive rewards)`: proportion of prompts yielding zero gradients due to uniformly positive rewards; and `zero-gradient-ratio (by all negative rewards)`: proportion of prompts yielding zero gradients due to uniformly negative rewards.

We visualize these dynamics in Figure 2 for the `Qwen2.5-Math-7B` model trained on the `DAPO-MATH-17K` dataset. Each mini-batch contains $M = 256$ prompts with $N = 8$ rollouts per prompt. The results reveal several concerning patterns:

**Low Overall Effectiveness:** The effective gradient ratio consistently remains below 60%, meaning that over 40% of sampled data fails to contribute to model updates—a significant waste of computational resources.

**Dynamic Training Phases:** The gradient dynamics exhibit three distinct phases:

- **Early Training (0-70 iterations, approximately the first epoch):** The model struggles with most tasks, leading to predominantly all-negative rewards (green line peaks near 95%). This results in minimal learning signals being generated.

- **Mid Training (70-600 iterations):** As the model improves, it begins solving some tasks while still failing others, creating the mixed outcomes necessary for effective gradients. The effective gradient ratio can maintain above 40% during this phase.
- **Late Training (600+ iterations):** Tasks become increasingly easy, leading to a rise in all-positive rewards (orange line increases to 40%). Simultaneously, challenging tasks still result in all-negative rewards (the green line fluctuates around 20%). As a result, the effective-gradient-ratio steadily decreases to about 20% by 1000 iterations.

We provide theoretical analysis toward understanding the above empirical observations in the next section.

### 3.2 THEORETICAL ANALYSIS

We model reward outcomes as Bernoulli random variables to analyze the exploration budget required.

**Definition 1** (Success Rate). *We define the success rate $p$ on a prompt $x$ as the probability that the model generates a correct response:* $p_i \equiv p(x_i) = \mathbb{E}_{y \sim \pi_\theta(\cdot|x_i)}[r(y|x_i)] = \Pr[r(y|x_i) = 1]$.

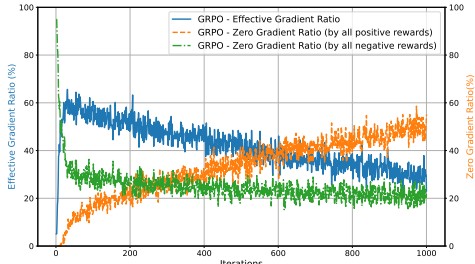

Figure 2: The ratio of effective gradients and zero gradients during training.

This formulation allows statistical analysis of stochastic gradients. For $N$ sampled responses $y_{i1}, \ldots, y_{iN}$ on a prompt $x_i$, the probability that both correct and incorrect samples are observed is:

$$\mathbb{P}(g_i \neq 0) = 1 - \mathbb{P}[\text{all rewards are the same}] = 1 - \mathbb{P}[\text{all rewards are 1's}] - \mathbb{P}[\text{all rewards are 0's}]$$
$$= 1 - p_i^N - (1 - p_i)^N.$$

This raises the question: how large must the sampling budget $N$ be to obtain a non-zero gradient? We answer this from two perspectives: high-probability guarantees and expected sample complexity.

**Proposition 1** (Exploration Budget). *Given a prompt with the success rate $p \in (0, 1)$, we have that*

- ***High probability bound:*** *For any $\alpha \in (0, 1)$, to ensure $\mathbb{P}(g_i \neq 0) \geq \alpha$, it suffices to take $N \gtrsim \frac{\ln(1-\alpha)}{\ln(\max\{p_i, 1-p_i\})}$.*

- ***Expected number of rollouts:*** *Let $N^{\text{first}}$ denote the number of independent rollouts required until $g_i \neq 0$ is achieved for the first time. Its expectation is:* $\mathbb{E}[N^{\text{first}}] = 1/p + 1/(1 - p) - 1$.

Please refer to Appendix D.1 for the proof. To illustrate, for example, if $p = 0.5$, we need 3 samples on average to obtain a non-zero gradient. For a hard task with $p = 0.01$, we require 100 samples, and to achieve a 90% chance of non-zero gradient, we would need 229 samples.

We show the theoretical predictions in Figure 3. We employ the `Qwen2.5-Math-7B-Instruct` model to generate 256 responses for 1,000 prompts from the `DAPO-Math-17K` dataset. Then we estimate $p$ and compute the minimal budget $N$ needed for $g_i \neq 0$ from the data. We exclude prompts that with empirical success rate of 0.0 or 1.0, because our exploration budget 256 is not sufficient. The results show that a typical budget of $N = 8$ only covers tasks with $p \in [0.1, 0.9]$. For tasks with $p \approx 0$ or $p \approx 1$, even increasing $N$ to 16 or 32 is insufficient. Overall, our analysis shows that the sampling budget required for meaningful gradients could be much larger than what is practically used.[1] This also helps explain the low effective gradient ratio observed in Figure 2.

Existing practices typically address this kind of insufficient exploration challenge in two ways:

- **Increasing the exploration budget uniformly.** This involves raising $N$—for example, from 8 to 16 or even 32—which could help address exploration on extremely hard or easy tasks and improve the effective gradient ratio. However, setting a very large value for $N$, such as $N = 100$, is often impractical due to prohibitive computational costs.

---

[1] We note that Proposition 1 addresses only whether the gradient is zero or non-zero, without considering the quality or informativeness of the gradient. Under this formulation, even relatively easy tasks may require substantial rollouts to obtain a non-zero gradient—a reasonable approach with unlimited compute, but impractical under finite computational budgets. In the following section, we take a more nuanced approach by incorporating gradient quality into our value formulation, which naturally supports allocating smaller exploration budgets to easier tasks when computation is finite.

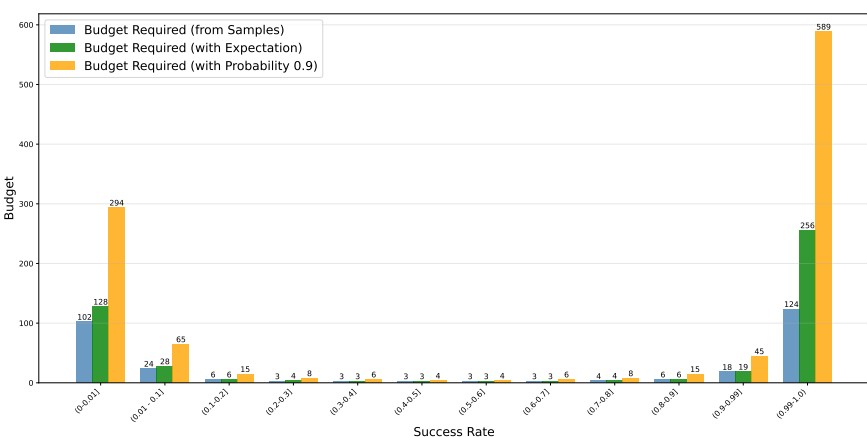

Figure 3: Exploration budget required to ensure non-zero gradients based on success rate. Note that success rates with in the same bins are grouped from real samples, which may not be symmetry, rendering the exploration budget may not be symmetry as the theory suggests.

- **Filtering hard and easy prompts.** Tasks that are too easy or too hard are dropped. This kind of approach is leveraged in (Team et al., 2025; Yu et al., 2025). However, as we have seen, the proportion of prompts yielding zero gradients due to all-negative rewards (the green line in Figure 2) is around 20% in late training, indicating many tasks are not yet fully solved. If we simply filter these prompts, we may close off a crucial source for RL, where meaningful learning often comes from converting failures into successes. That is, removing hard prompts deprives the model of opportunities to practice on challenging examples, limiting the information available to LLMs.

In this work, we favor addressing this issue by scaling exploration budgets, but recognize that this first approach presents a fundamental computation-exploration dilemma. This tension motivates our pursuit of a more principled solution for allocation of exploration budgets.

## 4 PROPOSED APPROACH: KNAPSACK-BASED RL

In this section, we introduce our approach to address the exploration-and-computation dilemma. Crucially, our goal is not to demand additional computational resources, as these are typically fixed by the user's constraints. Given these computational resources as a fixed pool, we aim to implement a centralized allocation strategy: assigning customized exploration budgets for each task.

The central technical question is: **given a fixed total budget, what is the optimal allocation for RL exploration?** Our key insight is that task difficulty alone does not dictate the optimal allocation—tasks also differ in their *value*. Easy tasks provide limited benefit, since correcting small mistakes leads to only incremental gains, whereas solving harder tasks can yield substantial improvements. This motivates reallocating budget from easier tasks to more challenging ones. In short, effective allocation must jointly account for both *exploration cost* and *learning value*.

We formalize the above idea as a constrained optimization problem:

$$\max_{N_1,\ldots,N_M} \sum_{i=1}^{M} \text{Value}(N_i, p_i) \tag{5}$$

$$\text{subject to} \sum_{i=1}^{M} N_i = N_{\text{total}}, \quad N_{\text{low}} \leq N_i \leq N_{\text{up}}, \quad N_i \in \mathbb{Z}^+,$$

where $N_i$ is the number of trajectories allocated to prompt $x_i$, and $p_i$ is the success rate. The bounds $N_{\text{low}}$ (e.g., 2) and $N_{\text{up}}$ (e.g., 128) allow to enforce coverage and prevent degenerate allocations. The total budget $N_{\text{total}}$ is usually set to $N \times M$ to match the homogenous allocation rule.

This optimization problem exactly matches the structure of a classical knapsack problem (Pisinger & Toth, 1998). Each prompt $x_i$ can be thought of as an item with a "weight" given by its allocated budget $N_i$ and a "value" of $\text{Value}(N_i, p_i)$. The objective is to choose budget allocations $N_i$ that maximize the total value while keeping the overall cost within the knapsack capacity $M \times N$.

## 4.1 FORMULATION OF TASK VALUE

In this section, we substantiate the above framework with the proposed idea. In contrast to the discussion in Section 3.2, which mainly focuses on acquiring a non-zero gradient, here we also consider the quality of gradient updates. For GRPO, we define the value of assigning $N_i$ exploration budget units to prompt $x_i$ as the expected improvement from one gradient descent step.

Specifically, we decompose the value over possible gradient events. When a zero-gradient event occurs (all samples positive or all negative), no learning occurs and the improvement is zero. When a non-zero gradient is obtained, the improvement depends on how informative that gradient is. This leads to our value formulation:

$$\text{Value}(N_i, p_i) = \text{ProbNonZeroGradient}(N_i, p_i) \times \text{InfoGain}(p_i),$$

where $\text{ProbNonZeroGradient}(N_i, p_i) = 1 - p_i^{N_i} - (1 - p_i)^{N_i}$ is the probability of obtaining a non-zero gradient, and $\text{InfoGain}(p_i)$ quantifies the expected improvement conditional on a non-zero gradient occurring. This formulation can also be extended to other algorithms; see Appendix E.

In this work, we define $\text{InfoGain}$ as a measure of the expected increase in success probability after a gradient update, while noting that alternative formulations could be explored in future work. Formally, let $p_i^t$ denote the success rate before the update and $p_i^{t+1}$ the rate after the update. We define

$$\text{InfoGain} = \Delta p_i = p_i^{t+1} - p_i^t.$$

Directly computing this requires access to the post-update success probability, which is intractable.

**Proposition 2.** *With the Taylor expansion, the* $\text{InfoGain}$ *can be approximated by* $\boxed{p_i(1 - p_i)^2}$.

Please refer to Appendix D.2 for detailed derivation. This functional form intuitively captures that the most valuable tasks are those that are challenging but not impossible (i.e., small but non-zero $p_i$).

## 4.2 PROPERTIES OF OPTIMAL ALLOCATION

With the value function defined, we can now analyze the structure of the optimal allocation it induces. For clarity, we restate our value function:

$$\text{Value}(N_i, p_i) := \left(1 - p_i^{N_i} - (1 - p_i)^{N_i}\right) p_i (1 - p_i)^2. \tag{6}$$

**Proposition 3.** *Fix any success probability* $p \in (0, 1)$*, we have the following results.*

- ***Monotonicity.*** $\text{Value}(N; p)$ *is non-decreasing in* $N$*; that is, adding more budget never decreases the value.*

- ***Diminishing returns.*** *The marginal gain* $\Delta \text{Value}(N; p) := \text{Value}(N + 1; p) - \text{Value}(N; p)$ *is strictly positive and strictly decreasing in* $N$*.*

- ***Hard-task bias.*** *For every* $N \geq 1$ *and every* $p \in (0, 0.5)$*, we have*

$$\text{Value}(N; p) > \text{Value}(N; 1 - p), \quad \text{and} \quad \Delta \text{Value}(N; p) > \Delta \text{Value}(N; 1 - p).$$

Please refer to Appendix D.3 for the proof. The properties of monotonicity and diminishing returns are crucial. They establish that our knapsack objective $\sum_i \text{Value}(N_i, p_i)$ is a sum of discrete concave functions. This structure is a classic hallmark of resource allocation problems that can be solved optimally with a simple greedy algorithm.

**Proposition 4** (Informal; see Appendix D.4 for a formal version)**.** *The optimal allocation* $N_i^*$ *for the knapsack problem in Equation* (5) *can be found via a greedy procedure. Starting with an initial allocation* $N_i = N_{low}$ *for all prompts, one unit of budget is iteratively assigned to the prompt* $j$ *with the highest current marginal gain,* $\Delta \text{Value}_j(N_j)$*, until the total budget is exhausted.*

**Implications for the optimal allocation.** Proposition 4, combined with the hard-task bias in Proposition 3, yields two key qualitative conclusions:

- The greedy algorithm naturally equalizes the marginal gains across all tasks, up to integer constraints. It allocates budget until the "value per rollout" is balanced across the batch, ensuring no budget is wasted on a task where it could be more productively used elsewhere.
- For any pair of mirror tasks with success probabilities $p$ and $1 - p$, the harder task has uniformly larger value and marginal value. Thus any optimal allocation must assign at least as much budget to the harder task.

This leads to a dynamic allocation strategy. With a small total budget, the allocation prioritizes "just-right" tasks (e.g., $p_i \in [0.2, 0.5]$) that offer the highest initial marginal gains. As the total budget increases, these tasks become "saturated" (their marginal gains diminish). The algorithm then pivots, allocating the remaining budget to very hard tasks (e.g., $p_i = 0.1$) to increase their chance of yielding a rare but valuable learning signal. Only with a sufficient large budget would it finally saturate even the easiest tasks.

### 4.3 Algorithm Implementation

In practice, the success rate $p_i$ is unknown and must be estimated from data. Directly estimating it with fresh rollouts at each step is computationally prohibitive. Instead, we adopt a simple heuristic from prior work (Team et al., 2025; An et al., 2025): using the success rates observed in the previous epoch as estimates $\widehat{p}_i$ for the current one. In our context, the first epoch may follow a homogeneous budget allocation rule, after which the proposed knapsack-based approach leverages the estimated success rates $\widehat{p}_i$ to guide allocation. Although this strategy introduces some delay and noise, it has proven empirically effective. More sophisticated online estimation techniques (e.g., logistic regression) that account for task correlations present promising directions for future improvement. Please refer to Appendix G.3 for more discussion and justification.

These estimated $\widehat{p}_i$ values are directly used to formulate the discrete constrained optimization problem (Equation 5), which can be solved in polynomial time using standard dynamic programming techniques or the greedy procedure (see Proposition 4). This step typically runs less than 1 second.

Overall, our knapsack-based exploration method integrates seamlessly into large-scale RL training pipelines with minimal modifications (see Listing 1 in the Appendix). Computationally, it adds negligible overhead. Algorithmically, it introduces no additional hyperparameters to tune and does not bias policy gradients. From a systems perspective, core components of inference (e.g., vLLM-based accelerated generation (Kwon et al., 2023)) and training (e.g., FSDP (Zhao et al., 2023) and Megatron (Shoeybi et al., 2019)) remain unchanged, ensuring full compatibility with existing infrastructure.

## 5 Experiments

### 5.1 Main Results

**Experiment Setting.** We implement Knapsack-RL and baseline methods using the large-scale RL training framework `Verl` (Sheng et al., 2025). Our primary focus is GRPO (Shao et al., 2024), a widely examined method, and we refer to our specific implementation as *Knapsack-GRPO*. Training utilizes the DAPO-Math-17K dataset (Yu et al., 2025), which comprises 17,917 prompts, each with a ground truth answer for verification.

We conduct experiments with both pre-trained and instruction-tuned models. The pre-trained models include Qwen3-4B-Base (Yang et al., 2025) and Qwen2.5-Math-7B (Yang et al., 2024). For instruction-tuned models, we utilize DeepSeek-R1-Distill-Qwen-1.5B (Guo et al., 2025) (abbreviated as DPSK-R1-Distill-1.5B) and Qwen3-4B-Instruct-2507 (Yang et al., 2025) (abbreviated as Qwen3-4B-Instruct). In each iteration, we employ a mini-batch size of $M = 256$ prompts and generate $N = 8$ rollouts. Our models are trained for 1,000 iterations. GRPO and Knapsack-GRPO require comparable total training time (approximately 1,400 GPU hours on A100s for Qwen2.5-Math-7B, or 1 day and 20 hours of wall-clock time with 32 GPUs). This is because the knapsack optimization adds negligible overhead, typically completing in under 1 second, while the per-iteration time remains dominated by response sampling and model training at approximately 130 seconds.

For evaluation, we follow (Luo et al., 2025b) and assess our method on several mathematical reasoning benchmarks: AIME, AMC, MATH, MINERVA, and OLYMPIAD Bench (OLYMPIAD for short). Given AIME's small sample size, we combine its 2024 and 2025 editions into a single dataset, hereafter referred to as AIME. Additionally, we include GPQA (Rein et al., 2023) as an out-of-domain evaluation, which tests scientific reasoning across physics, chemistry, and biology. All reported performance metrics are averaged over 16 generated responses.
We report the evaluation performance in Table 1, observing consistent improvements across all tested models after applying our RL training. Specifically, Knapsack-GRPO consistently outperforms GRPO. For instance, in terms of average performance, it improves by 3.8 points for DPSK-R1-Distill-1.5B compared to GRPO. On specific benchmarks, the improvements are even more significant: for example, 6.4 points on AIME for DPSK-R1-Distill-1.5B, 9.1 points on AMC for Qwen3-4B-Base, 5.5 points on GPQA for Qwen3-4B-Instruct, and 6.8 points on AMC for Qwen2.5-Math-7B.

Table 1: Evaluation performance (`avg@16`) comparison across different models and benchmarks.

| | AIME | AMC | MATH | MINERVA | OLYMPIAD | GPQA | Avg |
|---|---|---|---|---|---|---|---|
| DPSK-R1-Distill-1.5B | 25.3 | 62.1 | 81.4 | 25.8 | 41.7 | 39.1 | 42.9 |
| + GRPO | 27.6 | 71.1 | 84.0 | 27.6 | 46.4 | 36.7 | 45.9 |
| + Knapsack-GRPO | **34.0** | **75.1** | **86.7** | **28.5** | **49.7** | **40.3** | **49.7** |
| Qwen3-4B-Base | 6.6 | 29.9 | 48.0 | 19.4 | 23.1 | 26.4 | 22.9 |
| + GRPO | 20.7 | 56.9 | 80.6 | 31.9 | 44.9 | **46.6** | 43.2 |
| + Knapsack-GRPO | **20.8** | **66.0** | **81.0** | **35.7** | **46.2** | 45.5 | **45.1** |
| Qwen3-4B-Instruct | 47.7 | 82.5 | 92.4 | 35.4 | 61.6 | 43.0 | 58.6 |
| + GRPO | 47.0 | **84.9** | 92.5 | **41.8** | 61.8 | 54.4 | 59.2 |
| + Knapsack-GRPO | **48.2** | 83.1 | **92.5** | 38.2 | **63.5** | **59.9** | **61.9** |
| Qwen2.5-Math-7B | 12.3 | 41.0 | 61.2 | 11.8 | 26.1 | 22.0 | 26.7 |
| + GRPO | 23.9 | 70.6 | 81.7 | 33.6 | 41.9 | 40.8 | 45.2 |
| + Knapsack-GRPO | **24.3** | **77.4** | **83.9** | **34.5** | **44.1** | **43.8** | **47.5** |

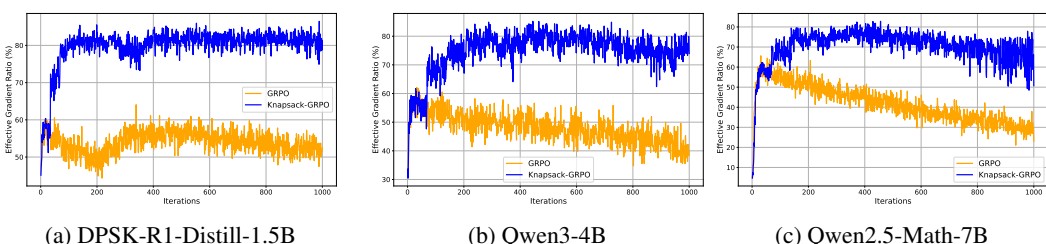

| (a) DPSK-R1-Distill-1.5B | (b) Qwen3-4B | (c) Qwen2.5-Math-7B |
|---|---|---|

Figure 5: Effective gradient ratio during training.

## 5.2 UNDERSTANDING KNAPSACK-BASED EXPLORATION

This section delves into understanding the superiority of knapsack-based exploration. We analyze its efficacy through gradient effectiveness and task status dynamics during training, focusing on the Qwen2.5-Math-7B model.

**Effective Gradient Ratio.** Figure 5 shows the effective gradient ratio during training, as defined in Equation (4). Knapsack-based budget allocation improves this ratio by approximately 20-40% across models. Unlike uniform allocation, the knapsack method avoids a clear decreasing trend. This stems from dynamically distributing exploration budgets, targeting tasks with mixed successful and failed trajectories. These observations partially explain Knapsack-GRPO's policy improvements.

**Task Transition Dynamics.** To understand our method's influence on learning, we analyze prompt evolution during training. We categorize prompts into five performance statuses based on success rate ($p_i$): `extremely-hard` ($p_i = 0$, all failures), `hard` ($0 < p_i \leq 0.2$), `medium` ($0.2 < p_i < 0.8$), `easy` ($0.8 \leq p_i < 1.0$), and `extremely-easy` ($p_i = 1.0$, all successes). Our anal-

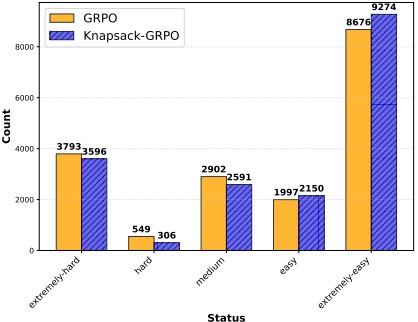

Figure 4: Distribution of sample statuses after training.

ysis covers two aspects: 1) prompt status transitions after training, and 2) final prompt status distribution.

Figure 6 visualizes the $5 \times 5$ transition matrix for Qwen2.5-Math-7B training, illustrating prompt category transitions. Knapsack-GRPO demonstrates superior efficiency in learning challenging tasks. Specifically, the self-absorption frequency for `extremely-hard` samples (prompts remaining in that status) is 43.4% for Knapsack-GRPO, notably lower than GRPO's 47.1%. Furthermore,

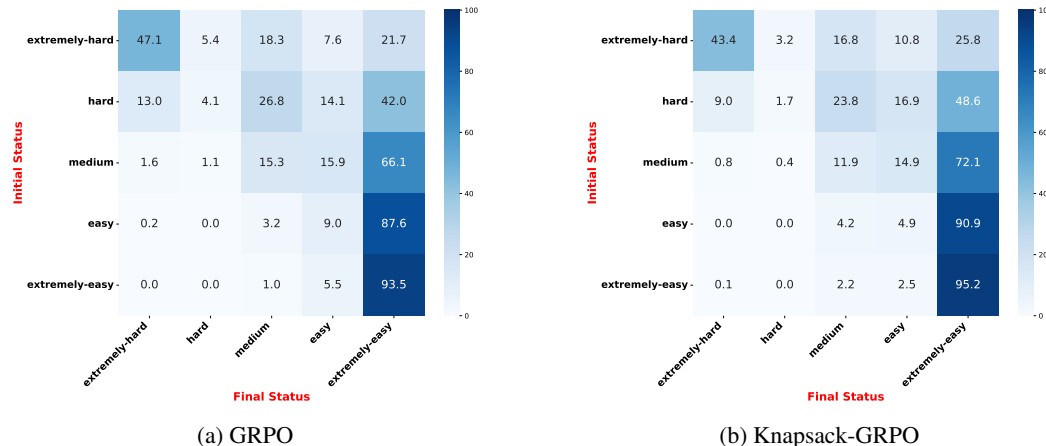

(a) GRPO                                (b) Knapsack-GRPO

Figure 6: Prompt transition matrices for Qwen2.5-Math-7B during training. The cell $(i, j)$ indicates the percentage of samples transitioning from status $i$ to status $j$.

Knapsack-GRPO shows a higher transition rate to `extremely-easy` tasks (last column in heatmap) than GRPO, indicating more effectively mastered samples.

We also examine the final distribution of prompt statuses after training, specifically by counting the training samples in each status, as depicted in Figure 4. Knapsack-GRPO has 3,596 `extremely-hard` tasks, less than GRPO's 3,793. This 197-task reduction suggests Knapsack-GRPO's dynamic budget allocation makes them more tractable. Consistent with observed transitions, Knapsack-GRPO yields 9,274 `extremely-easy` tasks, surpassing GRPO's 8,676. Despite these promising results, approximately 20% of prompts remain in the `extremely-hard` category even after 1,000 training iterations. We investigate if these are truly unsolvable: for Knapsack-GRPO, 577 of these challenging prompts recorded at least one positive trajectory during optimization, implying they are not inherently unsolvable. Future research could explore experience replay techniques to address these samples more effectively.

For a more detailed understanding of the Knapsack-GRPO exploration process, additional visualizations are included in Appendix G. These reveal that our knapsack-based method can assign up to **93 exploration budgets** to particular tasks, a dynamic allocation that is not computationally intractable with a uniform budget allocation approach.

## 5.3 EXPERIMENTS WITH DIFFERENT EXPLORATION BUDGETS

Finally, we conduct experiments with varying total exploration budgets to assess performance under different computational resource constraints. In contrast to previous experiments, which used a total budget of $N_{\text{total}} = 256 \times 8 = 2048$, here we explore scenarios with $N_{\text{total}} = 1024$ and $N_{\text{total}} = 4096$. For the vanilla GRPO, this corresponds to using $N = 4$ and $N = 16$, respectively. Note that the total budget parameter $N$ does not impact Knapsack-GRPO in the same way, given its distinct allocation strategy.

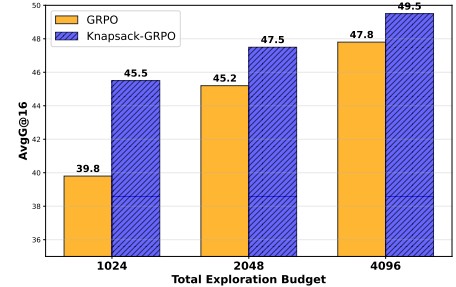

Figure 7: Performance comparison under different exploration budgets.

The results for the Qwen2.5-Math-7B model are shown in Figure 7. KnapSack-GRPO demonstrates clear advantages when computational resources are limited, improving performance from 39.8 to 45.5 in the low-budget setting, while maintaining its superiority even with larger rollouts. Notably, these results indicate that through more efficient exploration budget allocation, KnapSack-GRPO achieves performance levels that would require about 2x the computational resources for standard GRPO to match.

## 6 RELATED WORK

We discuss the relevant prior work in the main text and provide additional review in Appendix B.

**Resource Allocation.** Optimizing resource allocation has long been a key topic in operations research and systems engineering (Hurwicz, 1973; Katoh et al., 2013; Hussain et al., 2013; Maritan & Lee, 2017). Our work bridges the gap between these fields, RL, and LLM training, making it relevant for researchers across these areas. Traditionally, RL has focused on *single-task* settings, where computational budgets and resources do not need explicit distribution. In contrast, LLM training inherently involves *multi-task* scenarios, where a centralized resource pool must be effectively managed to handle the demands of multiple tasks simultaneously. In this context, we propose a knapsack-based optimization approach for allocating exploration budgets based on task difficulty. While our approach implicitly assumes that each rollout job takes roughly the same execution time, it is important to note that, from a test-time scaling perspective (Jaech et al., 2024), the rollout length may vary across tasks. To improve system efficiency, future work could incorporate finer-grained considerations, such as response length, into the allocation strategy. This is an important topic in RL training infrastructure design (Fu et al., 2025; Lu et al., 2025).

From a theoretical perspective, we note that while our work primarily adopts an offline knapsack formulation, where task information (e.g., task difficulty) is estimated beforehand, we also recognize the growing body of research on online exploration under knapsack constraints (Badanidiyuru et al., 2018; Chen et al., 2020; Brantley et al., 2020; Li et al., 2021; Liu et al., 2022). In the online setting, both allocation decisions and environmental information are learned jointly, with theoretical guarantees on regret. These studies predominantly focus on single-task settings, whereas our work addresses the multi-task learning. Developing theoretical foundations for resource allocation in this multi-task setting remains an open and promising area of investigation.

**Prompt selection and curriculum learning.** There is a line of research seeking to improve data efficiency through prompt selection and curriculum design (Lin et al., 2024; Zhang et al., 2024). For example, Chen et al. (2025b) used advantage estimates as a proxy for difficulty to construct curricula, while Sun et al. (2025a) employed perplexity as a difficulty metric. Additionally, Yu et al. (2025) presented the concept of "dynamic sampling" to address the issue of sparse gradients; however, it is crucial to clarify that their "sampling" refers to selecting prompts that yield effective gradients, rather than dynamically allocating exploration budgets. Recent work has advanced these ideas substantially. Bae et al. (2025) propose balanced online difficulty filtering that favors prompts yielding intermediate success rates. Qu et al. (2025) introduce a Bayesian model-predictive prompt selection method that predicts difficulty online without requiring full rollouts, enabling efficient prompt prioritization. Sun et al. (2025b) develop a difficulty-targeted online selection mechanism combined with rollout replay, framing difficulty as an adaptive quantity relative to the evolving policy.

Our work shares the principle that better assignment of compute to gain meaningful signals, but these studies are highly complementary to our work and differ in scope. Difficulty-aware prompt selection addresses *which prompts to train on*, implicitly assuming a roughly uniform exploration budget per selected prompt. In contrast, our knapsack-based formulation addresses *how much exploration to allocate to each prompt* under a global compute constraint. Rather than filtering to a subset of prompts, our method seeks to provide all tasks with exploration budgets sized appropriately to their training values. A promising avenue for future work is integrating our system-level, cross-task budget allocation with these online difficulty signals and within-task exploration strategies (see Appendix G.5 for preliminary experiments).

## 7 CONCLUSION

Motivated by the observation that RL agents require extensive exploration on challenging tasks to gather informative feedback and drive self-improvement, we investigate the problem of optimally allocating computational resources for exploration. We formulate this problem as a knapsack optimization, where each task-budget pair is treated as an item with an associated cost and value. This framework enables us to prioritize harder tasks, thereby yielding more effective gradients and leading to superior policy improvements. This comes at no additional computational cost, effectively offering a "free lunch". We view this work as an initial step toward unlocking RL's potential in LLM post-training through scaling exploration. Looking forward, moving beyond the straightforward stochastic rollout strategy considered here toward richer exploration methods and more structured allocation frameworks presents a promising avenue for future research.

## ETHICS AND REPRODUCIBILITY STATEMENT

This work primarily focuses on the algorithmic design for allocating exploration budgets within the context of RL training for language models. Our study is purely computational and does not involve human subjects, sensitive data, or any ethically contentious datasets. By enhancing training efficacy, our method aims to reduce overall computational costs and, consequently, mitigate the carbon footprint associated with large-scale model development.

To ensure full reproducibility of our findings, we provide comprehensive details regarding the training frameworks, hyper-parameters, and experimental settings in Appendix C and F. Furthermore, we are committed to publicly releasing our code, relevant datasets, and trained models for research purposes.

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

## A  USE OF LLMs

The drafting of this manuscript was enhanced through the use of a large language model, which assisted in grammatical refinement and the optimization of content organization.

## B  ADDITIONAL RELATED WORK

**Data heterogeneity.** A central challenge in RL for LLMs arises from the heterogeneity of training data. Prompts vary substantially in difficulty, leading to diverse reward distributions, and these distributions evolve throughout training, further complicating learning. Prior work has recognized this issue: for example, Li et al. (2024) observed substantial variations in reward distributions across prompts, which complicated stable gradient estimation. Their solution introduced refined baselines to reduce variance, thereby improving the *exploitation* stage of RL. Following this, many advanced policy optimization methods have been proposed (e.g., (Shao et al., 2024; Ahmadian et al., 2024; Yu et al., 2025)). We refer readers to surveys (Zhang et al., 2025a; Wang et al., 2025a) for a broader overview. By contrast, our work directly tackles the *exploration* challenge posed by heterogeneous data, focusing on how to allocate exploration resources more effectively to capture informative trajectories in the first place.

**Scaling RL.** Our method resonates with the principle of test-time scaling (Snell et al., 2024; Brown et al., 2024), which allocates additional computational resources (e.g., best-of-$N$ sampling, majority voting) to improve response quality. Similarly, our approach leverages extra compute to amplify exploration, thereby enhancing the quality of collected training signals. More broadly, our work aligns with recent efforts that scale computational budgets in post-training to unlock stronger downstream performance (Jaech et al., 2024; Liu et al., 2025).

**Exploration Strategy.** Our study focuses on simple on-policy exploration with independently sampled rollouts, primarily for its tractability and suitability for global budget reallocation. However, some difficult tasks may remain unsolved under independent sampling, motivating techniques that improve *how to explore within a fixed prompt*. Tree-based methods, inspired by Monte Carlo Tree Search (e.g., Silver et al., 2016), offer one such direction. Within LLMs, strategies such as state rollbacks and tree-structured reasoning have shown promise (Hou et al., 2025; Li et al., 2025). We view integrating such within-prompt exploration enhancements with our cross-prompt resource allocation as a fruitful direction.

Additionally, exploration quality is tightly coupled to policy entropy. To this end, Gao et al. (2025) introduce intrinsic-motivation-guided exploration signals that encourage diverse reasoning trajectories, while Cui et al. (2025) analyze the entropy collapse phenomenon in reasoning RL and propose regularization techniques to stabilize entropy. These developments are orthogonal to our contribution: such methods improve within-task exploration, whereas our work determines how to distribute exploration compute across tasks. Both dimensions can be jointly leveraged.

Finally, we notice that Yao et al. (2025) investigate resource allocation in the context of rejection sampling and RAFT (Dong et al., 2023), focusing on variance reduction. Their work differs from ours in two key ways: (i) it operates outside online RL, and (ii) it does not formulate exploration budget allocation as a multi-task knapsack problem balancing cost and learning value. Other studies such as Zhang et al. (2025b); Wang et al. (2025b) examine compute allocation during inference,

## C    IMPLEMENTATION

In this section, we provide more details in implementing Knapsack-based exploration.

**Handling Extreme Cases.** Our value function defined in Section 4.1 assigns a zero value to prompts with empirical success rates of 0 or 1, which would otherwise lead to zero budget allocation for these prompts. To prevent their complete exclusion and maintain coverage:

- For $\widehat{p}_i = 1.0$ (prompts always solved correctly), the estimate may be not accurate from history samples, so we allocate a small minimum budget (e.g., 2) to ensure they are still considered. This can be achieved by set $N_{\text{low}}$ in Equation (5).
- For $\widehat{p}_i = 0.0$ (prompts never solved correctly), we employ a fallback allocation strategy. We first estimate the total budget required for prompts with $p_i \in (0, 1]$ according to Proposition 1 and the above rule. Any remaining budget is subsequently distributed among extremely hard tasks. This strategy is particularly beneficial in later training stages where many prompts become easy, thus freeing up capacity to focus on hard tasks.

**Rollout Balancing.** In practice, the total number of trajectories ($M \times N$) is typically generated by $W$ parallel workers (where $W < M$), often leveraging efficient inference engines like vLLMs (Kwon et al., 2023). While a homogeneous allocation rule allows for a simple division of $M$ prompts among $W$ workers (each performing $N$ rollouts per prompt), our knapsack-based approach can lead to significant imbalance in allocated rollouts per prompt. This occurs because certain prompts may be allocated disproportionately large exploration budgets, creating an uneven workload and potentially leading to GPU idles and inefficient resource utilization.

To address this issue, we employ a simple rollout balancing strategy: we treat each allocated rollout for a prompt as an individual execution job. These execution jobs are then randomly dispatched to the available workers, with the inference engine generating one response per prompt. This approach is suitable for settings where prompts are not excessively long, thus not strictly requiring advanced techniques like prefix caching. For scenarios involving longer prompts, we would consider using the Karmarkar–Karp bin-packing algorithm (Karmarkar & Karp, 1982) to group prompts into approximately balanced batches based on their allocated budgets. Workers would then process these balanced groups of prompts, potentially utilizing prefix caching. This rollout balancing strategy is computationally inexpensive, typically taking less than 1 second in practice.

Listing 1: Python pseudo code implementation of Knapsack RL.

```python
def budget_allocation(batch, total_budget, **kwargs):
-    budget = np.full(len(batch), total_budget // len(batch['prompt']))
+    budget = knapsack(batch['status'], total_budget, **kwargs)
     indices = []
     for task_id, task_budget in enumerate(budget):
         if task_budget > 0:
             indices.extend([task_id] * task_budget)
     return batch.select_idxs(indices)

gen_batch = budget_allocation(batch, total_budget, **kwargs)
if rollout_balancing:
    indicies = np.random.shuffle(np.arange(len(batch['prompt'])))
    batch = batch.select_idxs(indicies)
batch = actor.generate_sequences(gen_batch)
batch = compute_rewards_and_advantages(batch)
train_dataset.update_status(batch)
actor.update(batch)
```

## D    PROOF

### D.1    PROOF OF PROPOSITION 1

*Proof of Proposition 1.* We prove both parts of the lemma.

**Part 1: High probability bound.** We want to find the minimum $N$ such that $\mathbb{P}(g_i \neq 0) \geq \alpha$ for a given $\alpha \in (0, 1)$. From the problem setup, we have:

$$\mathbb{P}(g_i \neq 0) = 1 - p_i^N - (1 - p_i)^N,$$

For the condition $\mathbb{P}(g_i \neq 0) \geq \alpha$ to hold, we require:

$$1 - p_i^N - (1 - p_i)^N \geq \alpha$$
$$p_i^N + (1 - p_i)^N \leq 1 - \alpha.$$

Let $q = \max\{p_i, 1 - p_i\}$. Since $p_i \in (0, 1)$, we have $q \geq \frac{1}{2}$. Without loss of generality, assume $p_i \geq \frac{1}{2}$, so $q = p_i$ and $1 - p_i \leq p_i$. The case $p_i < \frac{1}{2}$ follows by symmetry.

Since $(1 - p_i) \leq p_i$, we have $(1 - p_i)^N \leq p_i^N$ for $N \geq 1$. Therefore:

$$p_i^N + (1 - p_i)^N \leq 2p_i^N = 2q^N.$$

For large $N$, the term $q^N$ dominates $(1 - q)^N$ since $q > \frac{1}{2}$. More precisely, we have:

$$\lim_{N \to \infty} \frac{(1 - q)^N}{q^N} = \lim_{N \to \infty} \left(\frac{1 - q}{q}\right)^N = 0, \tag{7}$$

since $(1 - q)/q < 1$. Therefore, for sufficiently large $N$, the constraint (7) is dominated by the term $q^N$:

$$q^N \lesssim 1 - \alpha \quad \Longleftrightarrow \quad N \ln q \lesssim \ln(1 - \alpha). \tag{8}$$

Since $q < 1$, we have $\ln q < 0$, which gives:

$$\boxed{N \gtrsim \frac{\ln(1 - \alpha)}{\ln q} = \frac{\ln(1 - \alpha)}{\ln(\max\{p_i, 1 - p_i\})}.}$$

**Part 2: Expected number of rollouts.** Let $X_1, X_2, \ldots$ be i.i.d. Bernoulli random variables with $\Pr(X_i = 1) = p \in (0, 1)$, where 1 denotes "success" and 0 denotes "failure". Define

$$N^{\text{first}} \equiv N = \min\{n \geq 1 : \text{both 0 and 1 have appeared among } X_1, \ldots, X_n\}.$$

We compute $\mathbb{E}[N]$ by conditioning on the first trial $X_1$.

**Case 1: $X_1 = 1$ (probability $p$).** After the first success, we still need to wait until the first failure occurs. The waiting time for the first failure follows a geometric distribution with success probability $1 - p$, whose expectation is $1/(1 - p)$. Thus

$$\mathbb{E}[N \mid X_1 = 1] = 1 + \frac{1}{1 - p}.$$

**Case 2: $X_1 = 0$ (probability $1 - p$).** By symmetry, we wait for the first success; its waiting time has expectation $1/p$, so

$$\mathbb{E}[N \mid X_1 = 0] = 1 + \frac{1}{p}.$$

Applying the law of total expectation:

$$\mathbb{E}[N] = p\left(1 + \frac{1}{1 - p}\right) + (1 - p)\left(1 + \frac{1}{p}\right)$$
$$= \frac{1}{p} + \frac{1}{1 - p} - 1.$$

Hence, the expected number of rollouts until we first observe both a success and a failure is

$$\boxed{\mathbb{E}[N^{\text{first}}] = \frac{1}{p} + \frac{1}{1 - p} - 1.}$$

This completes the proof of the second part of Lemma 1. $\qquad \square$

## D.2 Proof of Proposition 2

*Proof of Proposition 2.* We make the following assumptions:

- The policy follows a softmax distribution: $p_k = \frac{\exp(z_k)}{\sum_{j=1}^{K} \exp(z_j)}$ for action $k$.

- The gradient update follows the policy gradient rule with advantage $A$:
$$z_k \leftarrow z_k + \eta \cdot A \cdot \mathbb{I}[k = y] \cdot \nabla_{z_k} \log p_y$$
where $\eta$ is the learning rate and $y$ is the chosen action.

- We assume unit learning rate ($\eta = 1$) and unit advantage ($A = 1$) for simplicity.

**Step 1: Taylor expansion.** For small parameter changes, the change in success probability can be approximated by:
$$\Delta p_y \approx \sum_{k=1}^{K} \frac{\partial p_y}{\partial z_k} \times \Delta z_k$$

**Step 2: Computing partial derivatives.** For the softmax probability $p_y = \frac{\exp(z_y)}{\sum_{j=1}^{K} \exp(z_j)}$, we have:
$$\frac{\partial p_y}{\partial z_y} = p_y(1 - p_y), \quad \text{and} \quad \frac{\partial p_y}{\partial z_k} = -p_y p_k, \quad \text{for } k \neq y.$$

**Step 3: Determining parameter updates.** Under the policy gradient update rule, we have:
$$\nabla_{z_k} \log p_y = \mathbb{I}[k = y] - p_k.$$
Therefore, the parameter updates are:
$$\Delta z_y = \mathbb{I}[y = y] - p_y = 1 - p_y,$$
$$\Delta z_k = \mathbb{I}[k = y] - p_k = 0 - p_k = -p_k, \quad \text{for } k \neq y$$

**Step 4: Computing InfoGain.** Substituting the partial derivatives and parameter updates:
$$\Delta p_y = \frac{\partial p_y}{\partial z_y} \Delta z_y + \sum_{k \neq y} \frac{\partial p_y}{\partial z_k} \Delta z_k$$
$$= p_y(1 - p_y) \cdot (1 - p_y) + \sum_{k \neq y} (-p_y p_k) \cdot (-p_k)$$
$$= p_y(1 - p_y)^2 + p_y \sum_{k \neq y} p_k^2$$

**Step 5: Simplification under first-order approximation.** For the first-order Taylor approximation to be accurate, we require small parameter updates. Under this condition, the cross-terms $\sum_{k \neq y} p_k^2$ are second-order in the update magnitude and can be neglected compared to the main term $p_y(1 - p_y)^2$.

Therefore, we obtain:
$$\text{InfoGain} \approx \boxed{p_y(1 - p_y)^2}.$$

$\square$

To validate this approximation, we conduct an empirical study with 100 actions, comparing the InfoGain computed through exact gradient updates against our theoretical approximation from Proposition 2. As shown in Figure 8, the two curves align closely across different success rates, demonstrating that our formula $p(1 - p)^2$ provides a reliable approximation for practical use.

### D.3 PROOF OF PROPOSITION 3

*Proof of Proposition 3.* To simplify notations, we write

$$\text{Value}(N_i, p_i) \equiv V(N_i; p_i) := \left(1 - p_i^{N_i} - (1 - p_i)^{N_i}\right) p_i (1 - p_i)^2$$

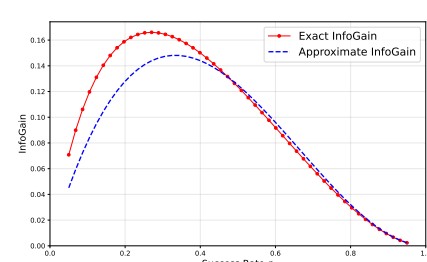

Figure 8: Comparison of exact InfoGain and approximate formula.

and
$$c(p) := p(1-p)^2 > 0, \qquad f(N;p) := 1 - p^N - (1-p)^N,$$
so that $V(N;p) = f(N;p)\,c(p)$.

**(1) Monotonicity.** We have
$$f(N+1;p) - f(N;p) = \left(1 - p^{N+1} - (1-p)^{N+1}\right) - \left(1 - p^N - (1-p)^N\right)$$
$$= p^N(1-p) + (1-p)^N p > 0,$$
since $p \in (0,1)$. Multiplying by $c(p) > 0$ shows
$$\Delta V(N;p) = V(N+1;p) - V(N;p) > 0,$$
so $V(N;p)$ is strictly increasing (hence non-decreasing) in $N$.

**(2) Diminishing returns.** Using the above expression,
$$\Delta V(N;p) = \left[p^N(1-p) + (1-p)^N p\right] c(p).$$
The second discrete difference is
$$\Delta^2 V(N;p) := \Delta V(N+1;p) - \Delta V(N;p)$$
$$= \left[p^{N+1}(1-p) + (1-p)^{N+1}p - p^N(1-p) - (1-p)^N p\right] c(p).$$
Factor the bracketed term:
$$p^{N+1}(1-p) - p^N(1-p) = p^N(1-p)(p-1) = -p^N(1-p)^2,$$
$$(1-p)^{N+1}p - (1-p)^N p = (1-p)^N p\big((1-p) - 1\big) = -(1-p)^N p^2.$$
Therefore,
$$\Delta^2 V(N;p) = \left[-p^N(1-p)^2 - (1-p)^N p^2\right] c(p) < 0,$$
since each factor is positive and the sum inside the brackets is negative. Hence $\Delta V(N;p)$ is strictly decreasing in $N$, establishing diminishing returns.

**(3) Hard-task bias.** Observe that $f(N;p)$ is symmetric in $p$ and $1-p$,
$$f(N;p) = 1 - p^N - (1-p)^N = f(N;1-p),$$
while the multiplicative factor $c(p)$ is not:
$$c(p) = p(1-p)^2, \qquad c(1-p) = (1-p)p^2.$$
Thus
$$\frac{V(N;p)}{V(N;1-p)} = \frac{f(N;p)c(p)}{f(N;1-p)c(1-p)} = \frac{c(p)}{c(1-p)} = \frac{p(1-p)^2}{(1-p)p^2} = \frac{1-p}{p}.$$
For $p \in (0,1/2)$, we have $(1-p)/p > 1$, so $V(N;p) > V(N;1-p)$.

An analogous argument applies to the marginal values. From the expression above,
$$\Delta V(N;p) = \left[p^N(1-p) + (1-p)^N p\right] c(p),$$
and the bracketed term is again symmetric in $p$ and $1-p$. Therefore
$$\frac{\Delta V(N;p)}{\Delta V(N;1-p)} = \frac{c(p)}{c(1-p)} = \frac{1-p}{p} > 1 \quad \text{for } p \in (0,1/2),$$
which implies $\Delta V(N;p) > \Delta V(N;1-p)$ for all $N \geq 1$. This completes the proof. $\square$

### D.4 PROOF OF PROPOSITION 4

We introduce the formal version of Proposition 4 below.

**Theorem 1** (Greedy optimal allocation). *Assume* $\text{Value}(N;p) \equiv V(N;p)$ *is given as in Equation* (6). *Furthermore, let* $\Delta V_i(N_i)$ *be the marginal value of $i$-th task. An optimal allocation $\{N_i^\star\}$ for Equation* (5) *can be obtained by the following greedy / water-filling procedure:*

- *Initialize* $N_i \leftarrow N_{\text{low}}$ *for all $i$.*

- *While $\sum_{i=1}^{M} N_i < N_{\text{total}}$:*
  - *Select an index*
  $$j \in \arg \max_{i:N_i < N_{\text{up}}} \Delta V_i(N_i),$$
  *i.e., the task with the largest current marginal gain that has not yet hit its upper bound.*
  - *Update $N_j \leftarrow N_j + 1$.*

*The resulting allocation $\{N_i\}$ is an optimal solution of Equation (5).*

*Proof of Theorem 1.* By Proposition 3(2), for each task $i$ the sequence $N \mapsto V(N; p_i)$ is discrete concave, so its marginal gains $\Delta V_i(N)$ are strictly decreasing in $N$. Consider an arbitrary feasible allocation $\{N_i\}$, and imagine constructing it by "adding" one trajectory at a time starting from the lower bounds.

Each trajectory addition to task $i$ corresponds to selecting one element from the decreasing sequence
$$\Delta V_i(N_{\text{low}}), \ \Delta V_i(N_{\text{low}} + 1), \ \ldots, \ \Delta V_i(N_{\text{up}} - 1).$$
The total objective value can therefore be written as the sum of exactly $N_{\text{total}} - M N_{\text{low}}$ chosen marginal gains across all tasks.

The greedy algorithm selects these marginal gains in non-increasing order, subject to the per-task capacity constraints (at most $N_{\text{up}} - N_{\text{low}}$ gains can be taken from each sequence). This is optimal because of a standard exchange argument: if some feasible allocation uses a marginal gain $\delta_a$ that is strictly smaller than another available marginal gain $\delta_b$ (which respects the same capacity constraints) that it did not use, then swapping $\delta_a$ for $\delta_b$ strictly increases the objective while preserving feasibility. By repeatedly applying such exchanges, any non-greedy allocation can be transformed into the greedy one without decreasing the objective, so the greedy allocation is optimal.

Formally, one can view the multiset of all candidate marginal gains
$$\left\{ \Delta V_i(N_{\text{low}}), \ldots, \Delta V_i(N_{\text{up}} - 1) \right\}_{i=1}^{M}$$
as a collection of sorted sequences; the greedy algorithm chooses the globally largest feasible entries, and any deviation from this choice admits an improving swap. This establishes the optimality of the greedy procedure. $\qquad \square$

# E  EXTENSIONS

In this work, we mainly focus on the widely used GRPO (Shao et al., 2024) algorithm to design the optimal allocation strategy. Here we discuss possible extensions for other RL algorithms by adapting the core framework while maintaining the same task value function structure:
$$\text{Value}(N_i, p_i) = \text{ProbNonZeroGradient}(N_i, p_i) \times \text{InfoGain}(p_i).$$
The key difference lies in how we compute $\text{ProbNonZeroGradient}(N_i, p_i)$ for different algorithms:

- **RLOO** (Ahmadian et al., 2024). RLOO's policy gradient estimator is equivalent to GRPO up to constants, thus we may not need fundamental changes. The probability of obtaining a non-zero gradient remains:
  $$\text{ProbNonZeroGradient}(N_i, p_i) = 1 - p_i^{N_i} - (1 - p_i)^{N_i}.$$

- **ReMax** (Li et al., 2024). ReMax leverages the reward of greedy response as baseline, rather than the averaged reward used in GRPO. In this setting, a gradient update occurs only when the sampled trajectory differs from the greedy response. If we denote the probability of the greedy response as $\alpha$, then the probability of sampling a trajectory different from the greedy response is $1 - \alpha$. The probability of obtaining a non-zero gradient with $N_i$ samples becomes:
  $$\text{ProbNonZeroGradient}(N_i, \alpha) = 1 - \alpha^{N_i}.$$
  This represents the probability that at least one of the $N_i$ sampled trajectories differs from the greedy response, thereby producing a gradient signal.

- **REINFORCE** (Williams, 1992). There is no baseline design in vanilla REINFORCE. We can directly calculate the $\mathrm{ProbNonZeroGradient}$ to account for the case where at least one trajectory receives a positive reward:

$$\mathrm{ProbNonZeroGradient}(N_i, p_i) = 1 - (1 - p_i)^{N_i}.$$

This formulation is simpler than GRPO since we only need to ensure at least one successful trajectory occurs, rather than balancing positive and negative samples.

The proposed framework's modularity allows for straightforward adaptation to other RL algorithms by: (1) identifying the algorithm's gradient computation mechanism, (2) determining conditions for non-zero gradients, (3) calculating the corresponding $\mathrm{ProbNonZeroGradient}$ function, and (4) maintaining the same $\mathrm{InfoGain}(p_i) = p_i(1 - p_i)^2$ formulation across algorithms. This demonstrates the general applicability of our value-based budget allocation approach beyond the specific GRPO implementation.

## F  EXPERIMENT DETAILS

Our experiments utilized the large-scale RL training framework `Verl`, specifically version 0.5.0. No modifications were made to the core training and inference code, with the exception of the advantage calculation, where values were clipped between -5 and 5. This was implemented because, as rollout responses were scaled, we observed their values could become significantly large in extreme cases, thus requiring this additional clipping for numerical stability. Additional implementation details on handling extreme cases and ensuring rollout efficiency are provided in Appendix C.

Following recommendations from (Yu et al., 2025), the learning rate was set to $10^{-6}$, with importance sampling clipping ratios (high/low) of $0.28$ and $0.2$, respectively. Neither KL nor entropy regularization was employed. Models were trained with a maximum sequence length of 4K tokens, with the exception of DPSK-R1-Distill-1.5B, which utilized 8K tokens to accommodate its typically longer chain-of-thought behaviors requiring more context.

For knapsack-based exploration, we use the success rate from rollouts in the last epoch, calculated empirically. Since our framework may assign a small budget (e.g., 2) to certain prompts, this can make the success rate estimation sensitive. To address this, we incorporate rollouts from earlier epochs, managing a buffer of the last 16 samples to smooth the success rate estimation.

For evaluation results reported during training, models were assessed every 10 training iterations using 16 generated responses. To manage evaluation time, 100 evaluation samples were randomly selected from benchmarks when the total number of samples exceeded this number.

For the final evaluation performance presented in Table 1, different maximum sequence lengths were used to prevent response truncation: 4K tokens for Qwen2.5-Math-7B, 8K tokens for Qwen3-4B and Qwen3-4B-Instruct, and 16K tokens for DPSK-R1-Distill-1.5B. Consequently, these results may not perfectly align with those reported in the training curves.

We conducted experiments using mixed computational resources: Qwen2.5-Math-7B was trained on 32×A100-80GB GPUs for 1 day and 20 hours; DPSK-R1-Distill-1.5B on 32×H20-96GB GPUs for 1 day and 10 hours; Qwen3-4B on 32×H20-96GB GPUs for 1 day and 5 hours; and Qwen3-4B-Instruct on the same hardware for 2 days and 16 hours. We note that the running time for Knapsack-GRPO is equivalent to GRPO, as its running time (usually less than 1 second) is computationally negligible.

## G  ADDITIONAL RESULTS

### G.1  VISUALIZATION OF EXPLORATION PROCESS

**Exploration Budgets.**  To illustrate the impact of knapsack-based exploration, we visualize the assigned exploration budgets. Specifically, we quantify the frequency with which different exploration budgets are allocated to individual prompts during the training of Qwen2.5-Math-7B. These results are presented in Figure 9. We observe that, even without introducing additional computational resources, our approach can dynamically assign up to 93 exploration budgets to certain tasks. This

level of dynamic, high-budget allocation is impractical to achieve under a conventional homogeneous budget allocation framework.

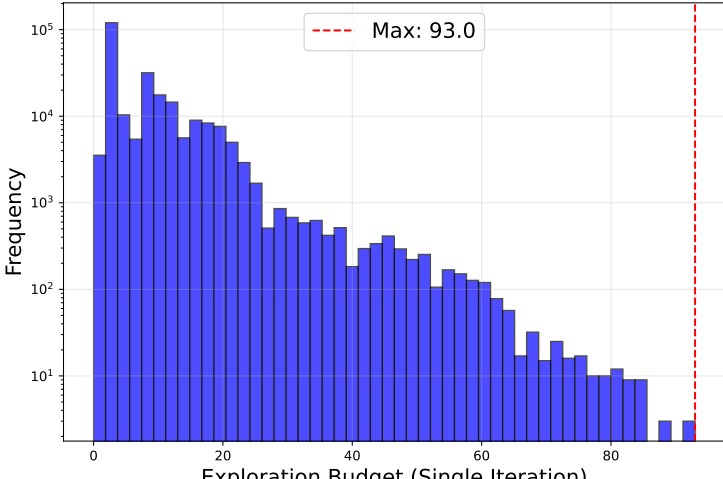

Figure 9: Distribution of exploration budgets allocated by knapsack-based exploration for Qwen2.5-Math-7B during training.

**Evolution of Prompts.** To illustrate the impact of exploration budgets on individual prompt learning dynamics, we track and visualize the learning trajectories of several randomly selected prompts from the training data in Figure 10. Each subplot corresponds to a unique prompt, identified by its index in the title. We observe that for several examples, our framework effectively allocates more exploration budget, leading to complete learning of the prompt (e.g., prompts in the first row, first column, and second row, first column). Conversely, some tasks remain highly challenging, where neither Knapsack-GRPO nor GRPO achieves satisfactory performance (e.g., the prompt in the third row, second column).

### G.2 TRAINING CURVES.

As references, the training curves for all models are displayed in Figures 11, 12, 13, and 14. Compared with the final results in Table 1, these plots further show that Knapsack-GRPO delivers a rapid performance improvement early in the training process. We also observe a few cases of performance degeneration, which points to the need for exploring more stable policy optimization techniques in future research.

### G.3 SUCCESS RATE ESTIMATION

We demonstrate the effectiveness of the success rate estimation strategy used in our work. Instead of relying on the success rate estimated from previous rollouts, we generate fresh rollouts (8 rollouts per prompt) at each step and estimate the success rate before each budget allocation.[2] As shown in Figure 15, we measure the error using the absolute difference between the success rate estimates. Initially, the error is around 0.25, but it decreases to 0.08 by the end of training. This trend is primarily due to the fact that the model parameters change slowly during training, allowing us to leverage past rollouts to more accurately estimate the current success rate. Given the small magnitude of this error, we expect it to have a minimal impact on the budget allocation.

Note that we also evaluate performance using this improved estimation strategy, which resulted in a final performance of 47.7, slightly better than the 47.5 achieved with our original strategy. However, this approach introduces a 2x increase in generation time, leading to slower training.

---

[2]While this approach eliminates the delay issue, it still cannot fully eliminate the error introduced by the finite number of samples.

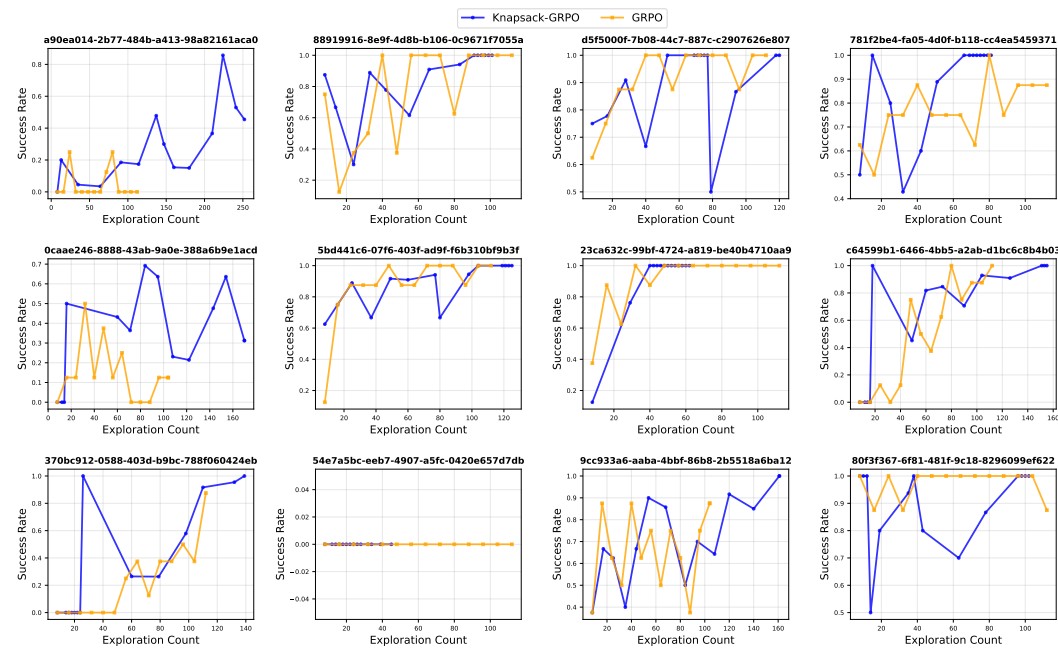

Figure 10: Learning dynamics of randomly selected prompts throughout training, comparing GRPO and Knapsack-GRPO. Each subplot shows the success rate evolution for a specific prompt.

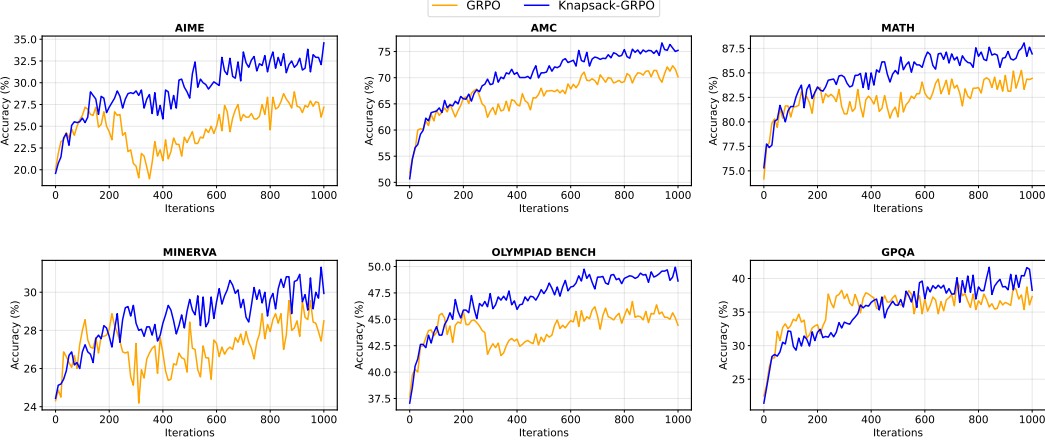

Figure 11: Evaluation performance of DPSK-R1-Distill-1.5B across training iterations.

We also have explored an online logistic regression method to predict success rates. Using a dataset of 10,000 samples, we train a 2-layer MLP on prompt embeddings from the Qwen3-0.6B-Embedding model. This predictor achieves an error of approximately 0.18 on a separate evaluation dataset of 256 samples, demonstrating the promise of this approach. Fully integrating this predictive model into the online training infrastructure remains a direction for future work.

### G.4    ABLATION STUDIES

**Without Fallback Strategy.**    In Appendix C, we introduced the *fallback strategy*, which reallocates excess exploration budgets from already-solved prompts to those that remain unsolved. This prevents a common failure mode: difficult prompts may otherwise receive too few resources, while easy prompts are oversampled.

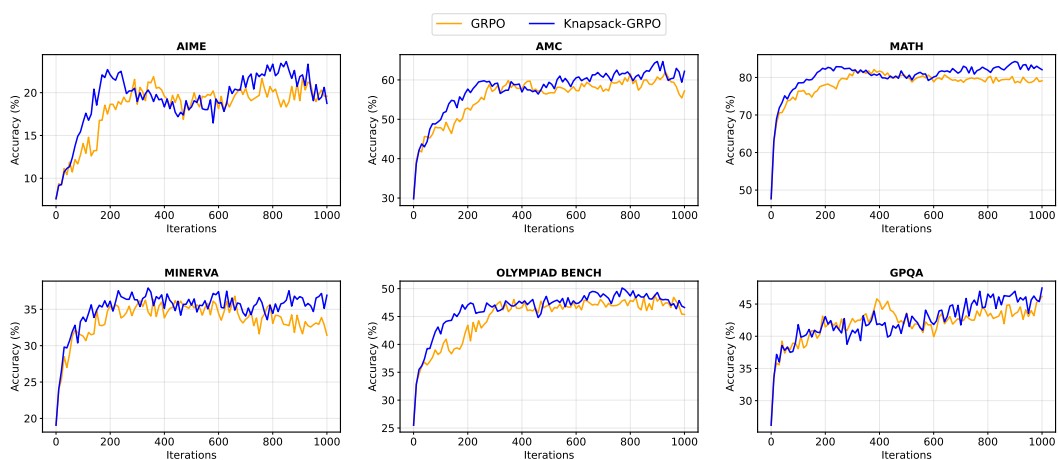

Figure 12: Evaluation performance of Qwen3-4B-Base across training iterations.

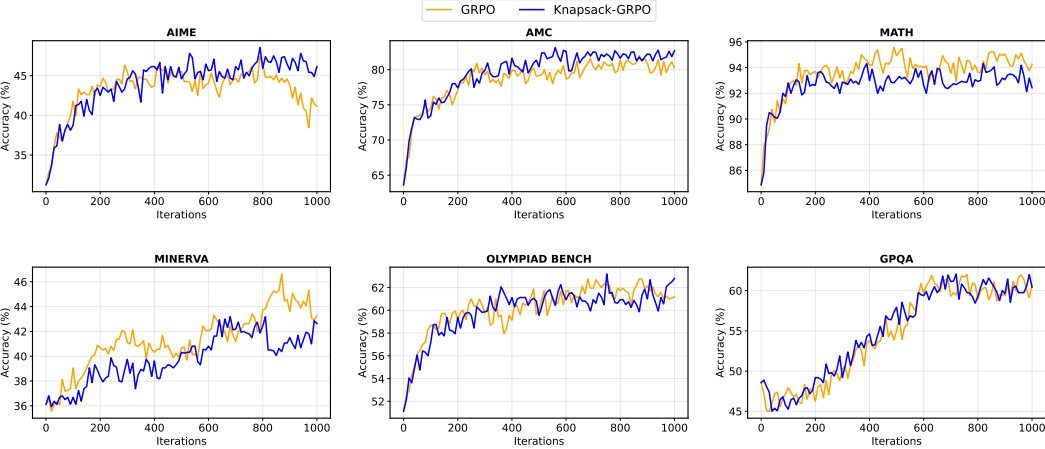

Figure 13: Evaluation performance of Qwen3-4B-Instruct across training iterations.

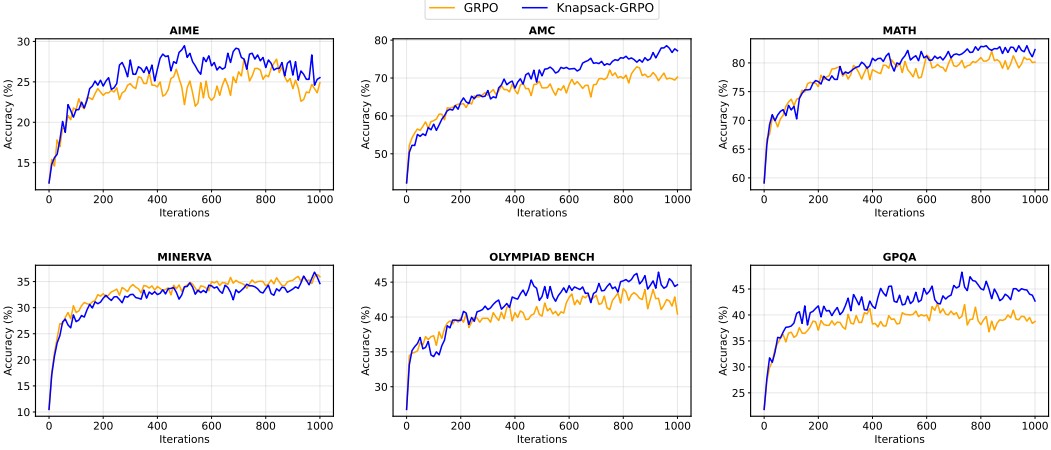

Figure 14: Evaluation performance of Qwen2.5-Math-7B across training iterations.

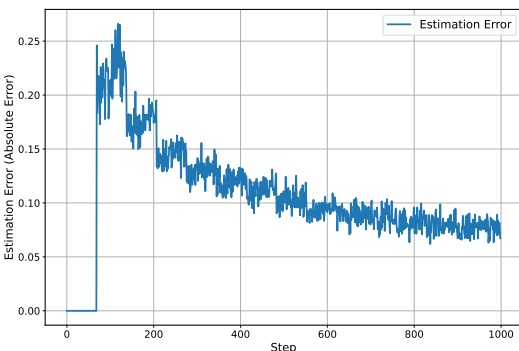

Figure 15: Error in success rate estimation over the course of training, measured using the total variation distance.

Table 2: Comparison of budget allocation with and without fallback strategy.

| Index | With Fallback Strategy | | | Without Fallback Strategy | | |
|---|---|---|---|---|---|---|
| | Success Rate | Cost | Assignment | Success Rate | Cost | Assignment |
| 1 | 0.0 | $\infty$ | 29 | 0.0 | $\infty$ | 2 |
| 2 | 0.9 | 22 | 23 | 0.9 | 22 | 50 |
| 3 | 1.0 | 0 | 2 | 1.0 | 0.0 | 2 |
| 4 | 1.0 | 0 | 2 | 1.0 | 0.0 | 2 |
| 5 | 1.0 | 0 | 2 | 1.0 | 0.0 | 2 |
| 6 | 1.0 | 0 | 2 | 1.0 | 0.0 | 2 |
| 7 | 1.0 | 0 | 2 | 1.0 | 0.0 | 2 |
| 8 | 1.0 | 0 | 2 | 1.0 | 0.0 | 2 |

A concrete example is shown in Table 2 with 8 prompts. Without the fallback strategy, the allocation assigns over 50 exploration units to a task with a success rate of 0.9, while the unsolved task (success rate 0.0) receives only 2 units. In contrast, with the fallback strategy, the unsolved task is assigned 29 units—substantially increasing its chance of making progress.

Empirically, this design proves crucial (Figure 16). In our experiments with the Qwen2.5-Math-7B model, removing the fallback strategy led to unstable training, large performance fluctuations on benchmarks such as AMC and OlympiadBench, and overall degraded results. This result suggests that neglecting challenging examples during training weakens the reinforcement signal, ultimately harming the model's ability to generalize.

**Low and Up Bounds.** Our framework incorporates safeguards in the form of hyper-parameters $N_{\text{low}}$ and $N_{\text{up}}$, as defined in Equation (5). $N_{\text{up}}$ is set to 128 primarily to facilitate faster computation of the knapsack optimization using dynamic programming; its specific value does not critically impact performance. Conversely, $N_{\text{low}}$ is set to 2 to prevent degenerate allocation scenarios, particularly when success rates might be inaccurate, as elaborated in Appendix C. We present ablation results for these bounds in Figure 17, which empirically support these design choices.

**Design of Task Value.** To demonstrate the necessity of both components in our task value formulation, we conduct a controlled ablation study. We first note that removing the probability term $\text{ProbNonZeroGradient}(N_i, p_i)$ would make the task value independent of the exploration budget $N_i$, rendering the optimization problem in Equation (5) underdetermined. In this degenerate case, uniform allocation trivially becomes an optimal solution, equivalent to vanilla GRPO. Therefore, we focus on examining the role of the $\text{InfoGain}(p_i)$ term by comparing our full model against a variant that uses only $\text{ProbNonZeroGradient}(N_i, p_i)$ as the task value.

Table 3 presents the optimal budget allocations for a synthetic task suite with uniformly distributed difficulties $p \in \{0.1, 0.2, \ldots, 0.9\}$ and a fixed total budget of $N_{\text{total}} = 72$ (equivalent to 8 rollouts

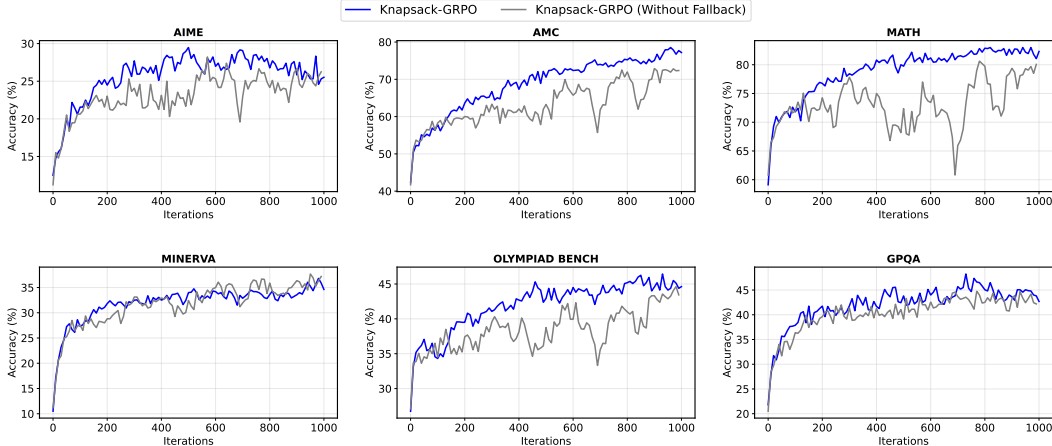

Figure 16: Effect of the fallback strategy. Without it, exploration budgets are disproportionately allocated to prompts with at least one successful trial, while unsolved tasks are largely ignored.

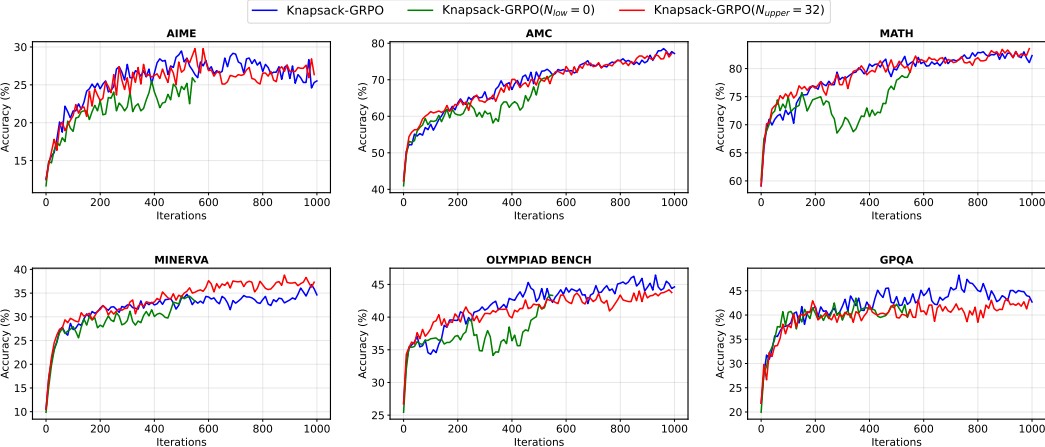

Figure 17: Ablation study on the impact of $N_{\text{low}}$ and $N_{\text{up}}$ constraints within the knapsack optimization framework.

per task under uniform allocation). The numbers indicate how many rollouts are allocated to each difficulty level under different value function designs.

Table 3: Budget allocation patterns under different value function designs. Numbers indicate rollouts allocated to tasks at each difficulty level ($p$).

| Method | Task Difficulty $p$ | | | | | | | | |
|---|---|---|---|---|---|---|---|---|---|
| | 0.1 | 0.2 | 0.3 | 0.4 | 0.5 | 0.6 | 0.7 | 0.8 | 0.9 |
| GRPO (Uniform) | 8 | 8 | 8 | 8 | 8 | 8 | 8 | 8 | 8 |
| Knapsack-GRPO | **14** | **12** | **9** | **7** | **7** | **7** | **7** | **6** | **3** |
| w/o InfoGain | 12 | 9 | 7 | 6 | 5 | 6 | 7 | 9 | 11 |

We see that without considering learning value, the optimal solution allocates budget solely to maximize non-zero gradient probability, which is given by $1 - p^N - (1-p)^N$. This objective produces a U-shaped allocation pattern that paradoxically favors both very easy ($p \geq 0.7$) and very hard ($p \leq 0.2$) tasks. We note that over-allocating to extremely easy tasks is suboptimal because they have limited learning value. By contrast, our full model incorporates $\text{InfoGain}(p_i) = p_i(1-p_i)^2$,

which concentrates budget on moderately-hard tasks ($p \in [0.1, 0.3]$) where the model struggles yet maintains sufficient success probability. This represents the optimal zone for learning—challenging enough to be informative, yet feasible enough to yield actionable gradients.

In addition, Table 4 validates these allocation patterns through downstream performance evaluation. We train Qwen2.5-Math-7B using each design variant and measure accuracy on six mathematical reasoning benchmarks. The results confirm that the InfoGain term is essential: removing it reduces average performance from 47.5% to 45.7% ($-1.8$ points), with consistent degradation across most benchmarks. This empirically demonstrates that effective budget allocation requires jointly considering both gradient availability (via ProbNonZeroGradient) and learning potential (via InfoGain).

Table 4: Evaluation performance (`avg@16`) on Qwen2.5-Math-7B across value function designs. Both NonZeroProb and InfoGain terms are necessary for optimal performance.

| Method | AIME | AMC | MATH | MINERVA | OLYMPIAD | GPQA | Average |
|---|---|---|---|---|---|---|---|
| Qwen2.5-Math-7B | 12.3 | 41.0 | 61.2 | 11.8 | 26.1 | 22.0 | 26.7 |
| + GRPO | 23.9 | 70.6 | 81.7 | 33.6 | 41.9 | 40.8 | 45.2 |
| + Knapsack-GRPO | **24.3** | **77.4** | 83.9 | **34.5** | **44.1** | **43.8** | **47.5** |
| w/o InfoGain | 22.7 | 71.8 | 81.5 | 32.4 | 41.9 | 47.0 | 45.7 |

**Results with Llama Models.** We provide additional results using the Llama-3.1-8B-Instruct model. For this model, we fine-tune with the `MATH` dataset (Hendrycks et al., 2021) instead of `DAPO-MATH-17K`, as the latter proves too challenging for Llama-3.1-8B-Instruct. Additionally, we disable off-policy updates, as we observe they cause training instability for this model over extended training iterations. The results are reported in Table 5. We observe that Knapsack-GRPO achieves the best overall performance, improving the average score from 18.7 (base model) to 24.5, outperforming standard GRPO (22.6) across most benchmarks, with particularly notable gains on AMC (29.1 vs. 23.6) and GPQA (38.5 vs. 34.9).

Table 5: Evaluation performance (`avg@16`) with Llama3.1-8B-Instruct model.

| | AIME | AMC | MATH | MINERVA | OLYMPIAD | GPQA | Avg |
|---|---|---|---|---|---|---|---|
| Llama3.1-8B-Instruct | 2.1 | 19.3 | 47.6 | 17.3 | 15.2 | 27.2 | 18.7 |
| + GRPO | **4.3** | 23.6 | 51.0 | 21.9 | 18.4 | 34.9 | 22.6 |
| + Knapsack-GRPO | 4.1 | **29.1** | **53.2** | **23.6** | **18.7** | **38.5** | **24.5** |

**Results with DeepScaleR Dataset** Beyond the `DAPO-MATH-17K` dataset, we also consider on the `DeepScaleR` training dataset (Luo et al., 2025b), which contains 40,315 training queries with verifiable ground truths. We fine-tune the Qwen2.5-Math-7B model on this dataset; results are shown in Table 6. We see that Knapsack-GRPO consistently outperforms standard GRPO, achieving an average score of 45.6 compared with 43.9. The improvements are particularly pronounced on AIME (28.4 vs. 25.0), MINERVA (37.1 vs. 34.7), OLYMPIAD (44.4 vs. 42.9), and GPQA (34.3 vs. 32.6), demonstrating the effectiveness of our approach across diverse mathematical reasoning benchmarks.

**Results with REINFORCE++.** Beyond GRPO, we also evaluate our exploration–budget allocation strategy with another RL algorithm, REINFORCE++ (Hu et al., 2025). As shown in Table 7, replacing the homogeneous allocation with our knapsack-based scheme consistently improves performance on most tasks (e.g., +6.6 on MINERVA and +4.5 on OLYMPIAD), while AIME remains comparable. Overall, the average score increases from 39.3 to 41.1.

Table 6: Evaluation performance (`avg@16`) with `DeepScaleR` dataset.

| | AIME | AMC | MATH | MINERVA | OLYMPIAD | GPQA | Avg |
|---|---|---|---|---|---|---|---|
| Qwen2.5-Math-7B | 12.3 | 41.0 | 61.2 | 11.8 | 26.1 | 22.0 | 26.7 |
| + GRPO | 25.0 | 63.9 | 83.2 | 34.7 | 42.9 | 32.6 | 43.9 |
| + Knapsack-GRPO | **28.4** | **62.7** | **84.1** | **37.1** | **44.4** | **34.3** | **45.6** |

Table 7: Evaluation performance (`avg@16`) with `REINFORCE++` training algorithm.

| | AIME | AMC | MATH | MINERVA | OLYMPIAD | GPQA | Avg |
|---|---|---|---|---|---|---|---|
| Qwen2.5-Math-7B | 12.3 | 41.0 | 61.2 | 11.8 | 26.1 | 22.0 | 26.7 |
| + REINFORCE++ | **20.2** | 62.4 | 76.5 | 25.3 | 32.5 | 38.3 | 39.3 |
| + Knapsack-REINFORCE++ | 18.9 | **63.2** | **78.9** | **31.9** | **37.0** | **39.1** | **41.1** |

### G.5 COMPARING WITH DYNAMIC SAMPLING IN DAPO

Dynamic sampling, a technique introduced in the DAPO paper (Yu et al., 2025), selects prompts with a mix of positive and negative rewards, filtering out those with exclusively positive or negative outcomes. This process is repeated until a target number of prompts is accumulated, a strategy that has been shown to be effective.

While effective, dynamic sampling operates on a different principle than our knapsack-based approach. Dynamic sampling aims to scale up effective **prompts**, while our method focuses on scaling up effective **responses**. Since these two approaches are parallel and can be combined, we conducted empirical studies to explore their synergy.

We evaluated the performance of these methods using two different metrics, as shown in the training curves in Figure 18 and Figure 19. Because dynamic sampling requires multiple exploration steps to accumulate enough effective prompts for a single gradient update, we can analyze performance in two ways:

- By exploration budget: Figure 18 shows performance relative to the total number of exploration iterations. This measures how effectively total computation budget is converted into performance gains. We found that dynamic sampling boosts GRPO's performance on benchmarks like AIME and OLYMPIAD, improving the average score from 45.2 to 46.2. When we combined dynamic sampling with our knapsack-based exploration, performance on the AMC benchmark improved significantly (from 69.8 to 73.0), resulting in a total performance of 46.5. This is slightly better than dynamic sampling alone but worse than our pure knapsack approach. We attribute this partially to the fact that knapsack-GRPO utilizes more gradient iterations, and therefore do not consider this a negative result.

- By gradient update iterations: Figure 19 displays performance against the number of gradient updates. This metric assesses the value of each gradient update. The results clearly show that effective gradients, whether from dynamic sampling or our knapsack-based exploration, lead to greater performance gains for the same number of update iterations, which validates the core motivation behind both techniques.

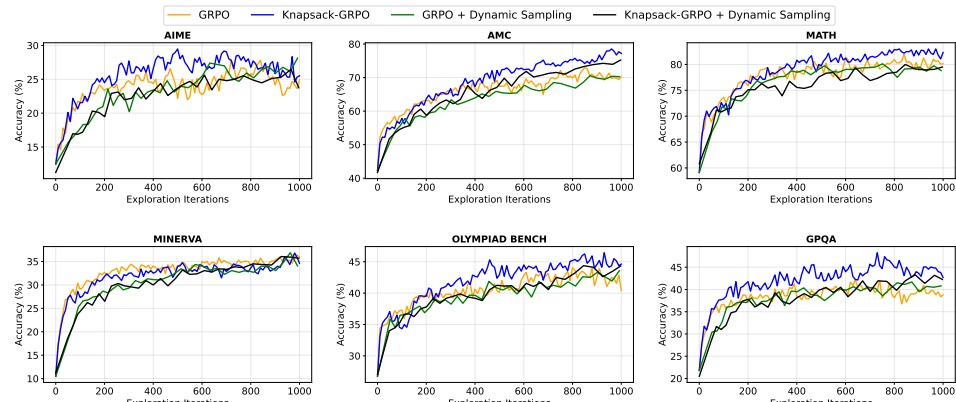

Figure 18: Performance of Qwen2.5-Math-7B relative to the number of exploration iterations, demonstrating how effectively the total computation budget is converted into performance gains.

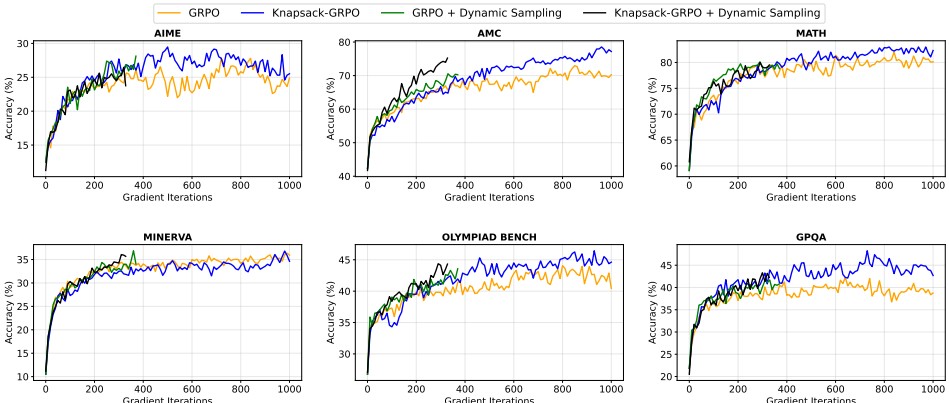

Figure 19: Performance of Qwen2.5-Math-7B as a function of the number of LLM gradient updates. This figure validates that effective gradients, derived from either dynamic sampling or the knapsack-based approach, lead to greater performance gains for the same number of updates.

