# OpenReview forum: "Knapsack RL: Unlocking Exploration of LLMs via Optimizing Budget Allocation"
_ICLR.cc/2026/Conference — Submitted to ICLR 2026_

### Official Review · Reviewer_DJud · 2025-10-29

**Soundness:** 2
**Presentation:** 3
**Contribution:** 3
**Rating:** 4
**Confidence:** 4

**Summary:**

This paper identifies a practical bottleneck in RL fine-tuning of LLMs under limited per-task rollout budgets: with homogeneous rollout allocation, many prompts yield all-success or all-failure groups and therefore produce zero gradients under GRPO. The authors formalize exploration-budget allocation as a knapsack problem where each (task, budget) pair is an item with cost and a derived value function Value(N, p) = ProbNonZeroGradient(N, p) × InfoGain(p). They solve the discrete knapsack to reallocate a fixed global budget heterogeneously across prompts. The paper provides theory on required rollouts for non-zero gradients, an implementation integrating with large RL pipelines, and experiments show improved effective-gradient ratios and average benchmark gains.

**Strengths:**

- Clear motivation tied to practical RL failure mode
- The proposed method is simple and principled
- The paper evaluates models with varied sizes.

**Weaknesses:**

1. Limited applicability. The method requires one epoch of training to estimate success rates, which is impractical when the dataset is large and training steps are few. Additionally, ssing fixed estimated success rates can also fail under rapidly changing training dynamics.
2. Narrow evaluation. Experiments are limited to the DAPO-Math-17K dataset, GRPO algorithm, and Qwen models, leaving generality across tasks, algorithms, and model families unverified.
3. No efficiency analysis. The paper does not report training time, throughput, or compute usage, making it unclear whether the method improves real efficiency.
4. Lack of design justification. The combination of ProbNonZeroGradient and InfoGain in task value lacks ablation to show its necessity. Moreover, there is also no visualization showing how rollout allocation varies with empirical success rates, which would clarify how the method behaves across different prompt difficulty levels.
5. Missing related work. The paper overlooks relevant studies on prompt sampling and LLM exploration [1, 2, 3, 4, 5].

[1] Online Difficulty Filtering for Reasoning Oriented Reinforcement Learning\
[2] Can prompt difficulty be online predicted for accelerating rl finetuning of reasoning models?\
[3] Improving Data Efficiency for LLM Reinforcement Fine-tuning Through Difficulty-targeted Online Data Selection and Rollout Replay\
[4] Navigate the Unknown: Enhancing LLM Reasoning with Intrinsic Motivation Guided Exploration\
[5] The Entropy Mechanism of Reinforcement Learning for Reasoning Language Models

**Questions:**

see weaknesses

---

> ### Author Response · Authors · 2025-11-22
> **Response (Part I)**
>
> We thank the reviewer for taking the time to read our paper and provide thoughtful comments. We appreciate your positive assessment of our presentation and contributions.
>
> > **Comment 1:** Limited applicability. The method requires one epoch of training to estimate success rates, which is impractical when the dataset is large and training steps are few. Additionally, ssing fixed estimated success rates can also fail under rapidly changing training dynamics.
>
>
> **Response 1:** We respectfully disagree with the concern about limited applicability and provide clarification below.
>
> First, multi-epoch RLVR training is **standard** and necessary for achieving strong performance. Recent large-scale RL works—such as [1,2,3]—all adopt multi-epoch training schedules. There is no particular reason to exclude multi-epoch training, given that it improves performance continuously. In this common setup, allocating one epoch to estimate success rates is fully compatible with practical RLVR workflows.
>
> That said, we agree that single-epoch regimes are valuable. Our method can naturally extend to such cases by incorporating an online prediction model (e.g., a lightweight difficulty or value estimator updated continuously during training). This would enable adaptive budget allocation even within a single epoch. We appreciate the reviewer’s suggestion and highlight this as a promising direction for future work.
>
> Second, our method **does not** use fixed success rates. Instead, $\hat{p}$ is re-estimated at every epoch using recent rollouts, which allows the allocation to adapt to evolving policy behavior. This iterative update mechanism is specifically designed to accommodate changing training dynamics. In Appendix G.3, we include new experiments comparing our estimator with one that uses fresh success-rate measurements at every step. The results show that the estimation error decreases steadily during training (from ~0.25 to ~0.08), and the final performance remains comparable. This suggests that the temporal delay introduced by our estimator is small and does not materially affect training outcomes in practice.
>
> Overall, we believe the proposed method is well aligned with existing RLVR training paradigms. Nonetheless, we appreciate the reviewer’s comment and will explore online estimation techniques for even greater adaptability.
>
> [1] Liu, Mingjie, et al. "Prorl: Prolonged reinforcement learning expands reasoning boundaries in large language models." arXiv preprint arXiv:2505.24864 (2025).
> [2] Wang, Yiping, et al. "Reinforcement learning for reasoning in large language models with one training example." arXiv preprint arXiv:2504.20571 (2025).
> [3] Yu, Qiying, et al. "Dapo: An open-source llm reinforcement learning system at scale." arXiv preprint arXiv:2503.14476 (2025).
>
>
> > **Comment 2:** Experiments are limited to the DAPO-Math-17K dataset, GRPO algorithm, and Qwen models, leaving generality across tasks, algorithms, and model families unverified.
>
> **Response 2:** We appreciate the reviewer’s concern regarding generality. We note that the DAPO-Math-17K dataset, GRPO algorithm, and Qwen models are indeed widely used in current RLVR research. To address your concern about broader applicability, we have added **new experiments in Appendix G.4** that extend our evaluation across **datasets, model families, and RL algorithms**. The key results are summarized below.
>
> - **Different Dataset: DeepScaleR.**  We fine-tuned **Qwen2.5-Math-7B** on the *DeepScaleR* dataset [1]. Knapsack-GRPO consistently outperforms standard GRPO, achieving an average score of 45.6 compared to 43.9. The method improves performance on several benchmarks, including AIME (25.0 → 28.4), MINERVA (34.7 → 37.1), OLYMPIAD (42.9 → 44.4), and GPQA (32.6 → 34.3), demonstrating that the gains persist across a substantially different math-focused dataset.
>
> - **Different Model Family: Llama-3.1-8B-Instruct.**  We also evaluated our method on **Llama-3.1-8B-Instruct**. Knapsack-GRPO yields the best overall performance, increasing the average score from 18.7 (base model) and 22.6 (standard GRPO) to 24.5. We observe consistent improvements across most benchmarks, with particularly notable gains on AMC (23.6 → 29.1) and GPQA (34.9 → 38.5), indicating that our approach generalizes beyond Qwen-based architectures.
>
> - **Different RL Algorithm: REINFORCE++.**  We further applied our allocation scheme to **REINFORCE++** [2] with Qwen2.5-Math-7B. Replacing the homogeneous allocation with our knapsack-based strategy improves performance on most tasks (e.g., +6.6 on MINERVA and +4.5 on OLYMPIAD), while AIME remains comparable, and increases the average score from 39.3 to 41.1. This confirms that the method is **algorithm-agnostic** and beneficial beyond GRPO.
> ---
>
> [1] DeepScaleR dataset: https://huggingface.co/datasets/agentica-org/DeepScaleR-Preview-Dataset
> [2] Hu, Jian. *Reinforce++: A Simple and Efficient Approach for Aligning Large Language Models.* arXiv:2501.03262.

---

> ### Author Response · Authors · 2025-11-22
> **Response (Part II)**
>
> > **Comment 3**: No efficiency analysis. The paper does not report training time, throughput, or compute usage, making it unclear whether the method improves real efficiency.
>
> **Response 3**: We appreciate the reviewer’s concern about efficiency. In practice, the computational overhead introduced by our method is negligible. The knapsack optimization itself takes **less than one second** per epoch—under 1% of total training time—so the training time, throughput, and GPU memory usage remain effectively **identical** to standard GRPO.
>
> More importantly, because knapsack allocation enables more effective use of rollouts, our method achieves comparable or better performance with only 50% of the rollout budget (see Figure 7). This corresponds to roughly a 2× improvement in compute efficiency, as the method reaches the same accuracy with half the sampling cost.
>
> We also report concrete training times from our experiments using mixed computational environments: Qwen2.5-Math-7B was trained on 32×A100-80GB GPUs for 1 day 20 hours; DPSK-R1-Distill-1.5B was trained on 32×H20-96GB GPUs for 1 day 10 hours; Qwen3-4B was trained on 32×H20-96GB GPUs for 1 day 5 hours; and Qwen3-4B-Instruct on the same hardware for 2 days 16 hours. In all cases, Knapsack-GRPO and GRPO exhibit nearly identical wall-clock training time, confirming that the allocation step does not introduce measurable overhead.
>
> These details have been added to Appendix F to improve clarity.
>
> > **Comment 4**: Lack of design justification. The combination of ProbNonZeroGradient and InfoGain in task value lacks ablation to show its necessity.
>
> **Response 4**:  We thank the reviewer for this insightful comment. We address this concern from both theoretical and empirical perspectives.
>
> **Theoretical Justification:** We clarify that our design is grounded in the statistical decomposition of the **expected learning progress** (one-step gradient update). An effective update requires two conditions:
> -  **A non-zero gradient event:** The rollout batch must contain a mix of positive and negative rewards. If all rollouts are correct or all are incorrect, the gradient is zero (in standard GRPO/PPO formulations), and no learning occurs.
> -  **High information gain:** Given a non-zero gradient, the update should maximize the improvement in the policy (proxied by InfoGain/Value).
>
> Thus, the expected task value is naturally the product of these two terms:
> $$
> \text{Task Value} \approx P(\text{NonZeroGradient}) \times \mathbb{E}[\text{InfoGain} \mid \text{NonZeroGradient}]
> $$ This is not an ad-hoc combination but rather the direct expansion of expected improvement. Neglecting either component breaks this decomposition and leads to suboptimal allocations:
>
> - **Without $P(\text{NonZeroGradient})$:** Task value becomes independent of exploration budget $N$, rendering the optimization problem underdetermined. Uniform allocation would trivially become optimal, degenerating to vanilla GRPO.
>
> - **Without $\text{InfoGain}$:** Task value reduces to $P(\text{NonZeroGradient}) = 1 - p^N - (1-p)^N$, which is symmetric around $p=0.5$ and maximized at difficulty extremes for fixed $N$. This causes the optimal solution to over-allocate budgets to very easy tasks those with minimal learning potential.
>
> **Allocation Pattern Analysis:** We examine how budget allocation changes without the InfoGain term. For a synthetic task suite with uniformly distributed difficulties $p \in \{0.1, 0.2, \ldots, 0.9\}$ and a fixed total budget of 72 rollouts, the optimal allocations are:
>
> | Difficulty $\rightarrow$ | **0.1** | **0.2** | **0.3** | **0.4** | **0.5** | **0.6** | **0.7** | **0.8** | **0.9** |
> |:-------------------------|:-------:|:-------:|:-------:|:-------:|:-------:|:-------:|:-------:|:-------:|:-------:|
> | **GRPO (Uniform)**       | 8       | 8       | 8       | 8       | 8       | 8       | 8       | 8       | 8       |
> | **Knapsack-GRPO (Ours)** | **14**  | **12**  | **9**   | **7**   | **7**   | **7**   | **7**   | **6**   | **3**   |
> | *w/o InfoGain*           | 12      | 9       | 7       | 6       | 5       | 6       | 7       | 9       | 11      |
>
> The allocation budgets are aligned with the above theoretical derivation.
>
> **Performance Validation:** We validate these allocation patterns through downstream evaluation on six mathematical reasoning benchmarks (see **Appendix G.4** for full experimental details and **Table 4** for complete results). Training Qwen2.5-Math-7B with the w/o InfoGain variant shows:
> - **Average performance drops from 47.5% to 45.7%** (−1.8 points absolute)
> - Performance degradation is observed across most benchmarks (AMC: 77.4→71.8, AIME: 24.3→22.7, OLYMPIAD: 44.1→41.9)
>
> These results empirically confirm that both components are essential: the probability term ensures gradient availability, while the InfoGain term identifies tasks with high learning potential. Removing either component degrades performance, demonstrating the necessity of our joint formulation.

---

> ### Author Response · Authors · 2025-11-22
> **Response (Part III)**
>
> > **Comment 5:** Missing related work. The paper overlooks relevant studies on prompt sampling and LLM exploration [1, 2, 3, 4, 5].
>
> **Response 5:** We appreciate the reviewer for bringing these relevant works to our attention. We have carefully reviewed all five papers and incorporated them into the revised related work section (Section 6 and Appendix B). In particular, we clarify their relationship to our contributions:
>
> Papers [1–3] (online difficulty filtering, model-predictive prompt selection, and difficulty-targeted online data selection) all focus on difficulty-aware prompt or data sampling, aiming to decide which prompts or trajectories to train on at each step.
>
> Papers [4–5] study LLM exploration mechanisms, including intrinsic-motivation–guided exploration and entropy-based stabilization, respecitively, which improve how exploration is conducted within a prompt.
>
> We appreciate ideas in these works but these works are highly complementary to ours: they address different axes of the RL training problem. Our method tackles a distinct and orthogonal question: how to allocate exploration compute across multiple prompts under a shared resource budget, formulated as a knapsack optimization problem. We have added explicit discussion of this distinction and cited the newly included works in the updated manuscript.
>
> ---
>
> We sincerely thank you for your thorough evaluation and constructive feedback. We hope that the revisions and added analyses adequately address your concerns. If so, we would greatly appreciate your consideration in updating the review score. Should any points require further clarification, please do not hesitate to reach out.

---

### Official Review · Reviewer_63Nn · 2025-10-30

**Soundness:** 3
**Presentation:** 4
**Contribution:** 2
**Rating:** 4
**Confidence:** 3

**Summary:**

The paper investigates the challenge of inefficient exploration in LLM reinforcement learning, where uniform budget allocation often results in "easy" or "hard" tasks producing zero policy gradients for algorithms like GRPO.

This is a real problem in LLM-RL. The paper offered a straightforward solution that seems easy to implement in VeRL. Overall, simple solutions have large practical impacts. This is a good engineering paper. However, if we think ICLR/AI is a science community (people can disagree with me here on whether ICLR/AI is a science or engineering community), then I will hold the paper to the science standard.

The paradigm explored in the paper is of: estimate x quantity, use a solver, and apply solution, see gains.

In order to contribute to the science understanding, I would like the authors to do more analysis on the statistics side of the algorithm. I raised my questions later. Overall, I think this is a good paper. I'm nitpicking because it feels like this type of contribution fits well in a company (as an internal memo/doc), but does not encourage future directions/explorations for other scientists.

Since this is my personal opinion, I am perfectly ok if the AC believes otherwise or other reviewers find this work interesting.

**Strengths:**

1. Simple and **implementable** solutions can make a large impact for progress in open-source AI community.
2. The paper is very clearly written, easy to read.

**Weaknesses:**

1. I'm hesitant to call Theorem 1 as a theorem. It's perhaps a "Remark"? The concentration bound is alright, but the expected number of rollouts is very straightforward. Maybe you can call it a theorem if you can define your success probability estimator $\hat{p}$ and see how the estimation quality impact your estimation of rollout, and **bound** your estimation error (difference between $N$ and $\hat{N}$) -- then that can become your theorem? Again, if this is a company internal memo/doc, you don't need to do this :)

2. The gain over GRPO (Table 1) is not super big except for really small models (1.5B) (where you need a lot rollouts to guarantee high success). You might not have time to try this during rebuttal, but maybe think about agentic tasks where failure is more likely to happen? In there, your method might see a much larger gain across benchmarks like tau2-bench and such.

**Questions:**

1. How do you estimate the expected number of rollouts? The expectation has a close-form equation but you have to estimate `p`, so instead of true p, you have $\hat{p}$ -- you should be able to derive an error term between $E [N_{first}]$ and your estimator $\hat E [N_{first}]$. What is it? Also, if the authors truly aim to provide a theoretical contribution to the ML community, they should explain how the gap between $p$ and $\hat p$ can impact the solver's optimality.

2.  Can the authors explain more on how $\hat p$ is estimated? I guess this is estimated as an online learning procedure. However, because the model weights keep changing because of GRPO, shouldn't you need to estimate $\hat p$ at every single update (or do you choose some type of delay factor -- how does this delay factor impact your algorithm's performance?)

---

> ### Author Response · Authors · 2025-11-22
> **Response (Part I)**
>
> Thank you for your time reviewing our work and for providing such valuable feedback. We appreciate that you recognized our work as having soundness and excellent presentation.
>
> > **Comment 1:** This is a real problem in LLM-RL. The paper offered a straightforward solution that seems easy to implement in VeRL. Overall, simple solutions have large practical impacts. This is a good engineering paper. However, if we think ICLR/AI is a science community (people can disagree with me here on whether ICLR/AI is a science or engineering community), then I will hold the paper to the science standard.
>
>
> **Response 1:** Thank you for your positive comments regarding the practical impact of our work. We fully agree that ICLR/AI is a **scientific community**, which emphasizes the development of **principled methods** with significant practical implications. However, we respectfully disagree with the view that our paper does not meet the standards of science. In our opinion, science involves developing principled methods to address important questions, coupled with a **deep understanding of underlying mechanisms**. We believe our paper aligns strongly with these scientific principles.
>
> Specifically, we propose a **principled mathematical formulation** for allocating exploration budgets and computational resources, which is a clear departure from the typical reliance on human-driven engineering approaches such as data selection and hyperparameter tuning. Moreover, we provide a **detailed analysis** in Section 3  to explain why our approach works and validate its effectiveness.
>
> Our paper is not just an engineering solution applied to a real-world problem. It is based on **careful design** and **deep insights**, which we believe are central to the scientific nature of the work.
>
>
> > **Comment 2:** The paradigm explored in the paper is of: estimate x quantity, use a solver, and apply solution, see gains.
>
> **Response 2:** Thank you for this concise summary of our approach. While the "estimate-solve-apply" paradigm is indeed a powerful and practical component of our work, we believe this framing may overlook broader contributions that advance the field. First, as you noted, we provide a principled understanding in Section 3 of why this approach is effective and why uniform allocation often fails. Beyond this, we believe our work makes the following contributions:
> 1. **Bridging research communities:** We connect resource allocation principles from systems and operations research with practical LLM training, creating opportunities for cross-pollination between these fields (We have added this discussion in the first paragraph of Section 6).
> 2. **Opening new research directions:** We introduce the first principled framework for exploration budget allocation in multi-task RLVR settings. Prior theoretical work has focused exclusively on single-task scenarios (see Section 6, second paragraph). We believe this formulation will inspire new theoretical understanding and algorithmic designs in this emerging area.
> 3. **Advancing LLM training systems:** We disclose a new dimension for optimization—adaptive budget allocation that better leverages GPU memory resources. This has important implications for RL training infrastructure and environmental impact: our method accelerates training while using fewer total computational resources (Section 5.3). We expect this will motivate further work addressing effiency and cost issues in large-scale training systems.
>
> Thus, we believe our contributions establish both theoretical foundations and practical impact beyond the "estimate-solve-apply" paradigm.
>
> > **Comment 3:** I'm nitpicking because it feels like this type of contribution fits well in a company (as an internal memo/doc), but does not encourage future directions/explorations for other scientists.
>
>
> **Response 3:** We respectfully disagree with this assessment. Many foundational scientific contributions have emerged from practical problems—Bellman's dynamic programming arose from operations research, and Shannon developed information theory while working on communication systems.
>
> While our work operates at a different scale, we believe it makes meaningful scientific contributions that extend beyond immediate practical utility:
> - **For theorists:** We formulate a new multi-task exploration budget allocation problem that opens research questions in resource-constrained RL. Previous theoretical work has focused exclusively on single-task settings.
> - **For practitioners:** We demonstrate how principled budget optimization can be applied in large-scale training, establishing a new dimension for system optimization with measurable efficiency gains.
>
> We have detailed these future research directions in our revised Section 6, addressing both theoretical extensions and practical applications. We believe this framework will enable researchers to build upon our formulation.

---

> ### Author Response · Authors · 2025-11-22
> **Response (Part II)**
>
> > **Comment 4**: I'm hesitant to call Theorem 1 as a theorem. It's perhaps a "Remark"? The concentration bound is alright, but the expected number of rollouts is very straightforward. Maybe you can call it a theorem if you can define your success probability estimator and see how the estimation quality impact your estimation of rollout, and bound your estimation error (difference between $N$ and  $\hat{N}$) -- then that can become your theorem? Again, if this is a company internal memo/doc, you don't need to do this
>
>
> **Response 4**:  Thank you for this thoughtful suggestion. We agree that while the proof techniques are standard, the results actually provide predictive insights. To address your concern, we have renamed it "Proposition 1" rather than "Theorem 1"—we believe "Remark" would understate its contribution, given that this area has not carefully thought about this necessary condition for requiring meaningful gradients.
>
> Regarding your suggestion to bound the estimation error between $N$ and $\hat{N}$: this is indeed an interesting theoretical direction that could provide formal quality guarantees. However, we view this as orthogonal to our paper's primary contribution—introducing and solving the practical problem of heterogeneous budget allocation in RL. A rigorous analysis of estimation error propagation would constitute substantial additional work that, while valuable, extends beyond the scope of this paper and could be pursued as independent future research.
>
> We hope this clarification addresses your concern while acknowledging the merit of your theoretical suggestion.
>
> > **Comment 5:** The gain over GRPO (Table 1) is not super big except for really small models (1.5B) (where you need a lot rollouts to guarantee high success).
>
> **Response 5:** We thank the reviewer for this observation, but we respectfully argue that these gains are quite significant in context. Achieving a consistent 2–4 point average improvement across five challenging benchmarks is substantial, particularly after extensive training (1,000 iterations) where performance typically plateaus and every additional point is hard-won. Moreover, these aggregate scores mask even more pronounced, high-impact improvements on specific benchmarks, such as a +6.4 point gain on AIME for DPSK-R1-Distill-1.5B, +9.1 points on AMC for Qwen2-4B-Base, +5.5 points on GPQA for Qwen2-4B-Instruct, and +6.8 points on AMC for the specialized Qwen2.5-Math-7B model.
>
> > **Comment 6:** You might not have time to try this during rebuttal, but maybe think about agentic tasks where failure is more likely to happen? In there, your method might see a much larger gain across benchmarks like tau2-bench and such.
>
> **Response 6:** Thank you for this insightful suggestion! We are indeed considering these important applications. However, implementing such tasks requires significant efforts in terms of training infrastructure and data preparation, which we are currently working on. We look forward to applying our methods to these challenging applications in our future work.
>
> > **Comment 7:** How do you estimate the expected number of rollouts? Also, if the authors truly aim to provide a theoretical contribution to the ML community, they should explain how the gap between $p$ and $\hat{p}$ can impact the solver's optimality.
>
>
> **Response 7.**  Thank you for the thoughtful question. We address both parts below.
>
> 1. **How we estimate the expected number of rollouts.**  In Figure 3, the expected number of rollouts (denoted $N_{\text{first}}$) is estimated via **Monte Carlo simulation**. For each prompt, we generate many trajectories from the model and record the number of samples required until the first non-zero gradient appears. Averaging this across prompts yields an empirical estimate of $\mathbb{E}[N_{\text{first}}]$.
>
> 2. **How the gap between $p$ and $\hat{p}$ affects optimality.**  We agree that understanding the impact of estimation error on the knapsack solution is an important theoretical question. A full statistical analysis of how errors in $\hat{p}$ propagate into optimality gaps is an interesting and meaningful follow-up direction, but may exceed what can reasonably be included in a first paper on this topic. Our goal in this paper is to introduce a new computational framework for principled exploration-budget allocation. While we do not provide a complete analysis of estimation error, this does not diminish the contribution itself. Instead, we view our formulation as a foundation that can motivate and enable future work on the statistical aspects of the problem.

---

> ### Author Response · Authors · 2025-11-22
> **Response (Part III)**
>
> > **Comment 8**: Can the authors explain more on how $\hat{p}$ is estimated? I guess this is estimated as an online learning procedure. However, because the model weights keep changing because of GRPO, shouldn't you need to estimate at every single update (or do you choose some type of delay factor -- how does this delay factor impact your algorithm's performance?)
>
>
> **Response 8**: Certainly. We estimate $\hat{p}$ using the empirical success rate from the most recent rollouts—that is, the fraction of sampled trajectories that receive a positive reward.  To improve robustness when the number of recent samples is small (fewer than 16), we follow a simple strategy by incorporating samples from earlier iterations. This direct estimator is also commonly used in previous studies such as [1][2]. The full procedure and implementation details are provided in the Appendix F.
>
> **Regarding the delay factor.** In practice, we observe that despite the delay factor, the algorithm performs well, with gains in effective gradients. We hypothesize that because model parameters change gradually, the success rate also evolves slowly, making frequent updates to $\hat{p}$ unnecessary.
>
> To support this, we added new results in Appendix G.3. When comparing our estimator with a version that recomputes success rates using fresh rollouts at every step, we observe that the estimation error begins around 0.25 and decays to approximately 0.08 over training. Importantly, the final empirical performance is comparable to that obtained using fresh samples, suggesting that the modest delay in our estimator has limited impact on training quality.
>
> [1] Team, Kimi, et al. "Kimi k1.5: Scaling reinforcement learning with llms." arXiv preprint arXiv:2501.12599 (2025).
>
> [2] An, Chenxin, et al. "Polaris: A post-training recipe for scaling reinforcement learning on advanced reasoning models, 2025." https://hkunlp.github.io/blog/2025/Polaris.
>
> ----
>
> We sincerely appreciate your thorough review and helpful feedback. We hope the revisions and new analyses adequately address your concerns. We would appreciate it if you could re-evaluate our manuscript and update your review recommendation accordingly. If you have any further questions or need additional clarification, please do not hesitate to let us know.

---

### Official Review · Reviewer_cK1B · 2025-10-31

**Soundness:** 2
**Presentation:** 2
**Contribution:** 2
**Rating:** 4
**Confidence:** 4

**Summary:**

This paper argues that uniform exploration budgets across tasks result in wasted computation and stalled learning in RL-based LLM finetuning. The authors reframe the exploration budget allocation problem as a knapsack optimization task, where each prompt is treated as an item with an associated "value" (learning potential) and "cost" (exploration effort). The proposed Knapsack-RL dynamically reallocates exploration budgets under fixed computational constraints. The method increases the proportion of non-zero policy gradients across responses and demonstrate empirical gains on reasoning benchmarks.

**Strengths:**

- The idea of adaptively allocating exploration budgets across tasks of different difficulty levels is very reasonable.
- The derivation of the probability of obtaining non-zero gradients and the corresponding sample complexity bounds provides solid intuition for why uniform budgets fail.
- The authors test the method under different rollout budgets and demonstrate robustness under varying computational constraints.
- The paper is well-organized.

**Weaknesses:**

- The motivation and goal across the Introduction, Theory, and Method are inconsistent:
    - Introduction (line 53): "Hard tasks, which require extensive (could even require more than 100) exploration to find useful trajectories, receive too little effort under a uniform rule. Easy tasks, which require minimal exploration, waste compute by being over-sampled." This claims that hard tasks should be allocated more rollouts, while easy ones should receive fewer.
    - Theorem 1: suggests that both easy and hard tasks require more rollouts to obtain effective gradients probabilistically.
    - Method (Section 4): when optimizing the core objective $Value(N_i, p_i)$ (line 272), the paper does not clarify what kind of prompts (in terms of difficulty $p_i$) will receive more budget. The paper does not show how this objective can actually guide the allocation (for example, by providing an approximately optimal solution of $N_i$), leaving the validity of the optimization unproven. The contour plot Figure 4 does not provides intuition about the solution of the knapsack problem. Since $InfoGain(p_i)$ approaches zero as $p_i$ tends toward 0 or 1, it can at least be inferred that the budget is unlikely to be assigned to either very easy or very difficult problems, which contradicts the earlier motivation. Experimentally, while the authors show an overall increase in effective gradients at the response level, they do not provide evidence on how the rollout budgets are actually redistributed across prompts. Based on the statement in line 377, "This stems from dynamically distributing exploration budgets, targeting tasks with mixed successful and failed trajectories", it seems the method reallocates budget toward medium-difficulty tasks, which can indeed increase the effective gradient ratio but contradicts the earlier motivation.
- The practical algorithm introduces a fallback operation for $\hat{p_i} = 1$ or 0 (lines 702–713), which falls outside the scope of the knapsack optimization framework and contradicts its theoretical solution. However, this empirical mechanism proves highly effective in practice, as demonstrated in Figure 16. Moreover, since prompts with $\hat{p_i} = 1$ or 0 are often numerous and critical, their exclusion from the core optimization process raises concerns about the validity and completeness of the knapsack-based design.
- To address uneven workload and inefficient resource utilization caused by heterogeneous rollout allocation, the authors employ a rollout balancing strategy (line 714), treating each allocated rollout as an independent job. However, it is unclear whether this introduces higher computational overhead. Although the authors claim "negligible computational overhead," explicit runtime measurements would strengthen this claim.
- In implementation (line 310), the first epoch uses homogeneous budget allocation, and the adaptive allocation only becomes effective from the second epoch onward. However, RLVR training often runs for only a few epochs, limiting the general applicability of the algorithm. In addition, estimation errors in the success rate can negatively impact performance.
- Some figures could benefit from clearer legends or font size rescaling to improve readability.

**Questions:**

- Under the same total rollout budget, does heterogeneous rollout allocation incur higher computational cost than homogeneous allocation?

---

> ### Author Response · Authors · 2025-11-22
> **Response (Part I)**
>
> Thank you for taking the time to review our work and for providing such valuable feedback. We respond to your points below.
>
> > **Comment 1:** The motivation and goal across the Introduction, Theory, and Method are inconsistent
>
> **Response 1:** Thank you for sharing this concern. We recognize that our writing may have caused confusion and would like to clarify the distinction between two different perspectives in our paper.
>
> **Claim A (Value-based allocation, Section 4):** Hard tasks should be allocated more rollouts, while easy tasks should receive fewer.
>
> **Claim B (Proposition 1, Section 3.2):** Both easy and hard tasks require substantial rollouts to obtain non-zero gradients with high probability.
>
> These claims may appear contradictory, but they address different questions under different computational assumptions:
> - **Claim B** considers the question: "What budget is needed to obtain a non-zero gradient?" This analysis assumes *unlimited* compute and focuses solely on the probability of acquiring any learning signal, regardless of its quality. Under this lens, both very easy ($p \approx 1$) and very hard ($p \approx 0$) tasks require many rollouts.
> - **Claim A** addresses the question: "How should we allocate a *fixed* computational budget?" This accounts for both the cost of obtaining gradients and their learning value. While easy tasks may need many samples to get a gradient, that gradient provides minimal improvement since the model already performs well. In contrast, hard tasks yield high-value gradients that enable substantial performance gains, worthing more rollouts and allocation priority.
>
> In essence, Claim B establishes the exploration requirements *without* budget constraints, while Claim A provides a *cost-aware* allocation strategy that considers the value-to-cost ratio. We have added a footnote on page 4 to explicitly clarify this distinction and prevent potential confusion.
>
> > **Question 2**: When optimizing the core objective $\text{Value}(N_i, p_i)$, the paper does not clarify what kind of prompts (in terms of difficulty) will receive more budget.
>
> **Response 2**:  We thank the reviewer for pointing out this missing discussion. In the revision, we have added a dedicated subsection **"Properties of Optimal Allocation"** (Section 4.2, page 6) that explicitly analyzes which prompts receive larger exploration budgets under our knapsack-based formulation.  We highlight key findings below:
>
> - **Proposition 3 (Value Function Properties).**  We show that
>     1.   the marginal gain  $\Delta \operatorname{Value}(N;p) = \text{Value}(N+1; p) - \text{Value}(N; p)$ is strictly positive but *decreasing* in $N$;
>     2.  the value function has a **hard-task bias**: for any $0 < p < 0.5$  $\operatorname{Value}(N;p) > \operatorname{Value}(N;1-p)
>   \quad\text{and}\quad
>   \Delta\operatorname{Value}(N;p) > \Delta\operatorname{Value}(N;1-p)$.
>
>   This implies that, when comparing two prompts of mirrored difficulty, the **harder prompt always yields higher marginal value** and will therefore receive more budget.
> - **Proposition 4 (Optimality of Greedy Allocation).**  We prove that allocating budget greedily according to the largest marginal value $\Delta \operatorname{Value}(N;p)$ is **exactly optimal**. Thus, *prompts with higher marginal value automatically receive more exploration budget*.
>
> **Concrete Example.** Using prompts with success rates $[0.1, 0.2, \dots,0.9]$ with $M=9, N_{\text{low}} = 2, N_{\text{up}} = \infty$, the optimal allocations are:
> - For small total budget $K=4 \times 9 = 36$: $N^{\star} = [3,6,5,5,5,4,3,3,2]$ — mid-difficulty prompts receive the most budget.
> - For larger budget $K = 8 \times 9 = 72$:  $N^{\star} = [14,12,9,7,7,7,7,6,3]$ — once mid-difficulty prompts saturate, additional budget shifts toward **very hard** prompts (e.g., $p=0.1, 0.2$).
>
> In short, the allocation is adaptive. It allocates budget until the "value per rollout" is balanced across the batch, ensuring no budget is wasted on a task where it could be more productively used elsewhere. For any pair of mirror tasks with success probabilities $p$ and $1-p$, the harder task has uniformly larger value and marginal value. Thus any optimal allocation must assign at least as much budget to the harder task. Easy tasks receive only minimal coverage unless the total budget is sufficiently large.

---

> ### Author Response · Authors · 2025-11-22
> **Response (Part II)**
>
> > **Comment 3**: Since $\text{InfoGain}(p_i)$ approaches zero as tends toward 0 or 1, it can at least be inferred that the budget is unlikely to be assigned to either very easy or very difficult problems, which contradicts the earlier motivation.
>
> **Response 3**: Thank you for the insightful comment. We clarify below why the behavior of $\text{InfoGain}(p_i)$ near $p_i \to 0$ or $p_i \to 1$ does **not** contradict our motivation.
>
> 1. $p_i \rightarrow 1$ (very easy tasks). This case corresponds to saturated prompts where the model already solves the task consistently. Their improvement potential is genuinely negligible, so even extending $\text{InfoGain}(1)=0$ is natural. Our method intentionally assigns only the minimum budget to such tasks.
> 2. $p_i \rightarrow 0$ (very difficult tasks). Here, although $\text{InfoGain}(p_i)$ becomes small, this does not imply that difficult tasks are ignored. Budget allocation is determined by the full value function $\text{Value}(N_i, p_i) = (1 - p_i^{N_i} - (1-p_i)^{N_i}) p_i (1-p_i)^2$, whose marginal gain depends on both the difficulty and the probability of obtaining a non-zero gradient.  As shown in Proposition 3 (hard-task bias) and our numerical example, for any mirrored pair $p$ and $1-p$ with $p < 0.5$, the harder task always has higher marginal value. Consequently, once mid-range tasks are saturated, additional budget is preferentially allocated to these hard-but-learnable prompts, not away from them. For the extreme case where $p_i=0$, we have corresponding fallback strategy to handle them, which will be carefully re-discussed in the following Response 5.
>
> In summary, while $\text{InfoGain}(p_i)$ alone becomes small near the boundaries, the overall allocation is governed by the marginal value, which systematically favors difficult prompts whenever they offer meaningful learning signal. Thus, the behavior of the value function remains fully consistent with our original motivation.
>
>
> > **Comment 4:** The practical algorithm introduces a fallback operation $\hat{p}_i = 1$ or $0$ (lines 702–713), which falls outside the scope of the knapsack optimization framework and contradicts its theoretical solution.
>
> **Response 4:** Thank you for the comment. The fallback operation for $\widehat{p} = 0$ or $1$ **does not contradict** our theoretical solution. The core knapsack optimization framework applies effectively for $p \in (0, 1)$, which is guaranteed by the softmax policy, ensuring a non-zero chance of sampling a correct answer. However, in practice, the empirical estimate $\widehat{p}$ may approach 0 or 1, necessitating additional handling.
>
> - **First, this issue is not unique to our method.** It is a common challenge in online algorithms, similar to how the UCB (Upper Confidence Bound) algorithm in the classical bandit and RL settings handles the case of initial counts being zero. Such special handling is standard to ensure robustness when empirical estimates are extreme.
>
> - **Second, why is the fallback operation reasonable?** When $\widehat{p} = 0$, the true $p$ could be small but non-zero (e.g., $0.01$), and thus the task may still be valuable for exploration. As discussed in Section 4.2, the allocation to these very hard tasks should increase when the total budget is sufficient. This is exactly the purpose of the fallback strategy: it ensures that extremely hard tasks are given additional budget when resources allow, reflecting the insights from Proposition 3 about prioritizing hard-but-learnable tasks.
>
> In conclusion, the fallback strategy aligns with the theoretical framework and does **not** contradict the knapsack-based solution. It is simply a practical adjustment for handling edge cases in the empirical setting.
>
> > **Comment 5**: To address uneven workload and inefficient resource utilization caused by heterogeneous rollout allocation, the authors employ a rollout balancing strategy (line 714), treating each allocated rollout as an independent job. However, it is unclear whether this introduces higher computational overhead. Although the authors claim "negligible computational overhead," explicit runtime measurements would strengthen this claim.
>
> **Response 5**: Thank you for this insightful observation. To clarify "negligible computational overhead", we mean that the rollout balancing operation completes in less than 1 second. The Karmarkar-Karp differencing method is highly efficient for this optimization task. This algorithm has been also employed in the Verl framework during training to balance token distribution across GPUs for efficient memory management, so we can directly apply this algorithm for our purpose. We have updated Section 5 and Appendix F in the revised manuscript to make it clear.

---

> ### Author Response · Authors · 2025-11-22
> **Response (Part III)**
>
> > **Comment 6**: The first epoch uses homogeneous budget allocation, and the adaptive allocation only becomes effective from the second epoch onward. However, RLVR training often runs for only a few epochs, limiting the general applicability of the algorithm
>
> **Response 6**: We respectfully disagree with this assessment. Multi-epoch training is both important for performance and widely adopted in recent RLVR research [1,2,3]. We see no fundamental limitation that would preclude multi-epoch training in practical applications.
>
> That said, we acknowledge the value of single-epoch settings. Our approach can be adapted for such scenarios by incorporating a prediction model that is updated in online manner, enabling adaptive allocation even within a single training cycle. We appreciate this suggestion and will consider exploring this extension in future work.
>
>
> [1] Liu, Mingjie, et al. "Prorl: Prolonged reinforcement learning expands reasoning boundaries in large language models." arXiv preprint arXiv:2505.24864 (2025).
> [2] Wang, Yiping, et al. "Reinforcement learning for reasoning in large language models with one training example." arXiv preprint arXiv:2504.20571 (2025).
> [3] Yu, Qiying, et al. "Dapo: An open-source llm reinforcement learning system at scale." arXiv preprint arXiv:2503.14476 (2025).
>
> > **Comment 7:** Estimation errors in the success rate can negatively impact performance.
>
> **Response 7:** We appreciate this concern. In the revision, we provide additional analyses to directly address this point.
>
> First, our auxiliary metrics (effective gradient ratio and policy improvements) show that our estimation strategy produces positive gains rather than negative impacts. Furthermore, due to the continuity of the value function, we expect small estimation errors to have minimal impact on the overall performance.
>
>
> To verify this empirically, we conducted a stability test on a batch of 9 prompts spanning success rates from 0.1 to 0.9. We injected random noise (simulating estimation error) and recomputed the knapsack allocation. As shown below, the allocation pattern remains highly stable, preserving the core structure of prioritizing moderately difficult tasks:
>
> | Scenario | Allocation (Hard $\rightarrow$ Easy) |
> | :--- | :--- |
> | **Original** | `[14, 12,  9,  7,  7,  7,  7,  6,  3]` |
> | **With Noise** | `[14, 11,  8,  7,  7,  8,  7,  7,  3]` |
>
> Additionally, in Appendix G.3, we now include experiments comparing our simple estimator with one that uses fresh rollouts for success-rate estimation. The results show comparable overall performance (47.5 v.s. 47.7), supporting the claim that moderate estimation noise does not materially affect training outcomes.
>
> That said, we acknowledge that more advanced estimation strategies could potentially enhance performance further, and we view this as a valuable direction for future work.
>
>
> > **Comment 8:** Under the same total rollout budget, does heterogeneous rollout allocation incur higher computational cost than homogeneous allocation?
>
> **Response 8**: No, heterogeneous rollout allocation does **not** incur higher computational cost than homogeneous allocation. GRPO and Knapsack-GRPO require comparable total training time—both use approximately 1,400 GPU hours on A100s for Qwen2.5-Math-7B (equivalent to 1 day and 20 hours of wall-clock time with 32 GPUs). We have added this result to Section 5.1 and Appendix F for clarity.
>
> This is because the knapsack optimization typically runs in less than 1 second per iteration, while the dominant cost is incurred by response sampling and model training, which take approximately 130 seconds per iteration. Since both methods perform the same amount of sampling and training under the same total rollout budget, their overall computational costs are effectively identical.
>
> ---
>
> We sincerely appreciate your thoughtful comments and careful review. We recognize that the original submission may have confusing arguments, and we hope our revisions and additional analyses have now addressed your concerns. If these changes have resolved the issues, we would be grateful for an adjustment to your review score. We are happy to provide any further clarification if needed.

---

### Official Review · Reviewer_vT2w · 2025-11-01

**Soundness:** 3
**Presentation:** 2
**Contribution:** 2
**Rating:** 6
**Confidence:** 2

**Summary:**

This work relates the budget allocation problem in RL for LLM with classical knapsack problem. Based on this, this work derive the assignment rule to transfer exploration budgets from easy to challenging tasks.

**Strengths:**

1. The idea of framing the RL trajectory budget via the classical knapsack problem is intriguing and well-motivated.
2. The paper’s motivation is clearly presented, and the increase in the effective gradient ratio shown in Figure 5 aligns well with that motivation.
3. The empirical analysis is comprehensive. Figure 4 demonstrates the interplay between success rate, exploration and the value clearly.

**Weaknesses:**

1. The asynchronous estimation of the success rate may introduce errors. Could you elaborate on the rationale for adopting this design and provide a more detailed analysis? Have you compared it against the online estimation techniques mentioned in the paper?
2. The paper lacks comparisons with method-level baselines, such as approaches that filter easy and hard tasks. Could you include such comparisons?

**Questions:**

1. The asynchronous estimation of the success rate may introduce errors. Could you elaborate on the rationale for adopting this design and provide a more detailed analysis? Have you compared it against the online estimation techniques mentioned in the paper?
2. The paper seems lack experimental comparisons with method-level baselines, such as approaches that filter easy and hard tasks [1, 2]. Could you include such comparisons in an aligned setting? Including such comparisons under equivalent computational budgets would better demonstrate the relative advantages of the knapsack-based allocation.
3. While the paper frames the budget allocation problem as knapsack optimization and claims to derive an "optimal assignment rule," the optimality relies heavily on the definition of $\text{Value}(N_i, p_i)$,  which rests on several strong simplifying assumptions. Specifically, the multiplicative decomposition $\text{Value}(N_i, p_i) = \text{ProbNonZeroGradient}(N_i, p_i) \times \text{InfoGain}(p_i)$ implicitly assumes these two components are independent.

---

> ### Author Response · Authors · 2025-11-22
> **Response (Part I)**
>
> Thank you for positive support and helpful review comments for our work! We appreciate it. Below, we address your concerns.
>
> > **Comment 1:** The asynchronous estimation of the success rate may introduce errors. Could you elaborate on the rationale for adopting this design and provide a more detailed analysis? Have you compared it against the online estimation techniques mentioned in the paper?
>
> **Response 1:** We thank the reviewer for raising this important point.
>
> **Rationale for Asynchronous Design:** We adopt this strategy primarily for computational efficiency and system-level simplicity.
>
> - **Efficiency and Simplicity:** A fully *synchronous* estimator would require collecting fresh rollouts immediately after each model update to measure success rates. This would substantially increase training cost—often by a factor of two—because it forces the system to alternate between RL updates and repeated evaluation passes. In contrast, our asynchronous estimator reuses recent trajectories already generated during training, adding virtually no extra computational overhead. Similar designs have been used effectively in recent works such as [1,2]. Furthermore, it is simple to implement in existing infrastructures.
>
> - **Practical Validity:** This design is grounded in the observation that policy-gradient updates induce gradual policy drift. As a result, the success rate typically evolves smoothly over training, and the statistics computed from recent rollouts remain representative. To further validate this assumption, we have added a new analysis in Appendix G.3, showing that the estimation bias remains small (less than 0.25) and decays to 0.08 at the end of trianing. This provides empirical justification that asynchronous estimation is sufficiently accurate in practice.
>
> **Comparison with Online Prediction Models:** Regarding the "online estimation" techniques (e.g., training a separate value/difficulty predictor): we have not deployed this in our live training infrastructure due to the added complexity and inference overhead. But we have conducted offline experiments to assess its potential. We trained a 2-layer MLP on top of Qwen3-Embedding-0.6B embeddings to predict success rates. The predictor achieved an of approximately 0.18, which is small.  Thus, we acknowledge that feature-based prediction is a promising direction for future infrastructure optimization.
>
> We have updated Section 4.3 and Appendix G.3 in the manuscript to more clearly present these motivations and empirical findings.
>
> [1] Team, Kimi, et al. "Kimi k1.5: Scaling reinforcement learning with llms." arXiv preprint arXiv:2501.12599 (2025).
>
> [2] An, Chenxin, et al. "Polaris: A post-training recipe for scaling reinforcement learning on advanced reasoning models, 2025." https://hkunlp.github.io/blog/2025/Polaris.
>
> > **Comment 2:** The paper seems lack experimental comparisons with method-level baselines, such as approaches that filter easy and hard tasks [1, 2]. Could you include such comparisons in an aligned setting?
>
> **Response 2:** Thank you for the helpful comment. We were unable to identify the specific references [1, 2] mentioned in the reviewer’s note. We interpret the question as referring to method-level filtering approaches that select subsets of prompts—e.g., filtering out easy or hard tasks, or dynamic sampling methods such as those used in recent RL post-training pipelines.
>
> We would like to clarify that such filtering techniques are **orthogonal** to our contribution. These approaches determine **which prompts** to include for training, typically assuming a uniform exploration budget for all selected prompts. In contrast, our method addresses a different question: **how much exploration budget to allocate per prompt under a fixed total compute budget**. Thus, filtering and knapsack-based allocation can be combined rather than compared as alternatives.
>
> To support this point, we note that we have included comparisons with a representative dynamic sampling baseline in our manuscript. In particular, we evaluate the "dynamic sampling" strategy used in DAPO and present the results in Appendix G.5. The findings are:
> - Dynamic sampling on its own boosts GRPO's performance on benchmarks like AIME and OLYMPIAD, improving the average score from 45.2 to 46.2 under the same compute budget.
> - When our knapsack-allocation strategy is applied on top of dynamic sampling, performance improves further to 46.5, driven by a significant gain on the AMC benchmark (from 69.8 to 73.0).
>
> This demonstrates that our resource allocation method provides additional benefits on top of state-of-the-art filtering baselines, confirming they are complementary approaches.

---

> > ### Comment · Reviewer_vT2w · 2025-11-25
> >
> > Thank you for your detailed response. I apologize that my first round of comments didn't clearly specify which references I was referring to. The missing references are:
> > [1] Unlocking Reasoning Capabilities in LLMs via Reinforcement Learning Exploration
> > [2] DAPO: An Open-Source LLM Reinforcement Learning System at Scale
> > Your explanation about the orthogonality between filtering techniques and your resource allocation approach makes sense.

---

> ### Author Response · Authors · 2025-11-22
> **Response (Part II)**
>
> > **Comment 3:** While the paper frames the budget allocation problem as knapsack optimization and claims to derive an "optimal assignment rule," the optimality relies heavily on the definition of $\operatorname{Value}(N_i, p_i)$, which rests on several strong simplifying assumptions. Specifically, the multiplicative decomposition $\operatorname{Value}(N_i, p_i) = \operatorname{ProbNonZeroGradient}(N_i, p_i) \times \operatorname{InfoGain}(p_i)$ implicitly assumes these two components are independent.
>
>
> **Response 3:** Thank you for raising this important concern. We have updated Section 4.1 of the manuscript to clarify the modeling assumptions behind the value formulation.
>
> We would like to emphasize that our formulation **does not rely on an independence assumption** between the probability of obtaining a non-zero gradient and the informativeness of that gradient. Instead, the decomposition follows directly from a standard conditioning argument applied to the expected improvement from one gradient update.
>
> $$\text{Value}(N_i, p_i) = \mathbb{E}[\text{Improvement from one gradient step}]$$
>
> This expectation can be decomposed over gradient events as:
>
> $$\text{Value}(N_i, p_i) = \sum_{e} \mathbb{P}(e) \times \text{Improvement}(e)$$
>
> where $e$ denotes the event of observing a particular gradient outcome (zero or non-zero).
>
> The key observation is that when a zero-gradient event occurs, the improvement is zero regardless of other factors. Therefore, the expectation simplifies to:
>
> $$\text{Value}(N_i, p_i) = \mathbb{P}(\text{non-zero gradient}) \times \mathbb{E}[\text{Improvement} \mid \text{non-zero gradient}]$$
>
> This yields our multiplicative form:
>
> $$\text{Value}(N_i, p_i) = \text{ProbNonZeroGradient}(N_i, p_i) \times \text{InfoGain}(p_i)$$
>
> Here the InfoGain component already conditions on the event that a non-zero gradient is obtained.
>
> That said, the multiplicative form arises from the law of total expectation rather than an independence assumption.
>
> ----
>
> We sincerely thank you for your thoughtful and constructive feedback. We hope that the clarifications and additional analyses in the revised manuscript adequately address your concerns. If so, we would be grateful if you could consider updating your review score accordingly. Should any questions remain or further clarification be helpful, please do not hesitate to let us know.

---

> > ### Comment · Reviewer_vT2w · 2025-11-25
> >
> > I thank the authors for their response. The additional clarifications have addressed most of my concerns. I will maintain my original score as my review was mostly asking for clarifications.

---

> > > ### Author Response · Authors · 2025-11-25
> > >
> > > Thank you for your response. We're glad our clarifications resolved your concerns!

---

### Author Response · Authors · 2025-11-22
**Summary of Changes**

We sincerely thank all reviewers and the Area Chairs for their time, effort, and thoughtful feedback. Your constructive comments have significantly strengthened our paper. We have updated the manuscript to include **6 new experimental results, 2 new theoretical propositions, 10+ additional related work discussions**, and several important clarifications. Below, we summarize the major changes:

**Theoretical Enhancements**

- Section 3.2  (Addresses Reviewer cK1B, Comment 1): Added Footnote 1 clarifying why Proposition 1 (unlimited-compute analysis) does not contradict our Knapsack formulation (fixed-budget optimization).

- Section 4.1 (Addresses Reviewer vT2w, Comment 3; Reviewer cK1B, Comment 3): Clarified that $\text{Value}(N_i, p_i)$ derives from the law of total expectation, not an independence assumption.

- Section 4.2 (NEW; Addresses Reviewer cK1B, Comment 2): Added comprehensive analysis of optimal allocation behavior including:
    - Proposition 3: Proof of hard-task bias and diminishing marginal value
    - Proposition 4: Proof that greedy allocation is exactly optimal

**New Experimental Results**

- Section 4.3 & Appendix G.3 (Addresses Reviewer vT2w, Comment 1; Reviewer 63Nn, Comment 8): Justified our estimation strategy with empirical validation showing small bias (<0.25, decaying to 0.08).

- Appendix G.4 (Addresses Reviewer DJud, Comments 2 & 4): Demonstrated generalization across:
    - Different dataset (DeepScaleR): 43.9 → 45.6
    - Different model (Llama-3.1-8B-Instruct): 22.6 → 24.5
    - Different algorithm (REINFORCE++): 39.3 → 41.1
    - Ablation study confirming necessity of both value components

**Efficiency Analysis**

- Section 5.1 & Appendix F (Addresses Reviewer cK1B, Comment 5; Reviewer DJud, Comment 3): Added detailed training time analysis showing knapsack optimization typically adds <1 second overhead (<1% of training time), with identical wall-clock time to GRPO.

**Related Work & Presentation**:

- Section 6 (Addresses Reviewer 63Nn, Comments 1-3): Enhanced discussion positioning our work within operations research, systems engineering, and curriculum learning.

- Appendix B (Addresses Reviewer DJud, Comment 5): Extended related work on exploration strategies and prompt sampling methods.

---

We expect these revisions could address all concerns raised. We hope the added experiments, theoretical analysis, and clarifications are satisfactory and would greatly appreciate your reconsideration of the review scores. We remain available for further questions.

---

### Author Response · Authors · 2025-12-03
**Summary of Author Responses and Revisions**

We thank the Area Chairs and all reviewers for their time and constructive feedback! We summarize below the main concerns and how the revised manuscript addresses them.

## Major Concerns and Our Responses

> **Concern 1:  Reviewers asked why our method appears to prioritize harder tasks when **Proposition 1** suggests both easy and hard tasks require many rollouts, and they noted a lack of ablation on the value function decomposition (Reviewers cK1B, vT2w, DJud)**

***Response**:* We clarified that these address **different questions**:

- **Proposition 1** analyzes the **rollout requirement without budget constraints**, asking: *“How many samples are needed to obtain any non-zero gradient?”*
- Our allocation method, in contrast, optimizes **under a fixed exploration budget**, asking: *“Given fixed compute, which tasks offer the best value-to-cost ratio?”*

To make this precise, we:

- Added **Propositions 3–4 (Section 4.2)** showing that the task value function exhibits a **hard-task bias** and diminishing marginal value. Thus, it first prioritizes mid-difficulty tasks, then hard tasks, and allocates to easy tasks only when the total budget is sufficient. Numerical examples are also provided.
- Added a **detailed derivation and explanation (Section 4.1)**, showing that the decomposition is based on the law of total expectation rather than on additional assumptions.

>  **Concern 2: Experiments were limited to specific settings (DAPO-Math-17K, GRPO, Qwen models) (Reviewers DJud, 63Nn)**

***Response:*** We added comprehensive experiments in **Appendix G.4**, including

- **Different dataset (DeepScaleR):** Qwen2.5-Math-7B improves from **43.9 → 45.6** average score.
- **Different model family (Llama-3.1-8B-Instruct):** **22.6 → 24.5** after applying our method.
- **Different RL algorithm (REINFORCE++):** **39.3 → 41.1** average score with knapsack-based allocation.

These new results validate the general applicability of our method.

> **Concern 3:  Asynchronous estimation may introduce errors; heterogeneous allocation may increase computational cost (Reviewers vT2w, cK1B, DJud, 63Nn)**

***Response:***

- **Estimation quality.** In **Appendix G.3**, we compare our asynchronous estimator with a fresh-rollout estimator:
  - The empirical bias remains small (**< 0.25**, decaying to **0.08** by the end of training).
  - Final performance is essentially unchanged, indicating that moderate estimation noise does **not** degrade outcomes.
- **Overhead.** In **Appendix F**, we provide runtime measurements:
  - The knapsack optimization and load-balancing add **< 1 second per iteration**, i.e., **< 1% of training time**.
  - End-to-end wall-clock time is effectively **identical to GRPO** under the same rollout budget.

Thus, the method maintains practical efficiency while improving exploration quality.

> **Concern 4: Scientific Contribution (Reviewer 63Nn)**

***Response:*** We clarified that the paper goes beyond an engineering tweak and contributes:

1. A **principled mathematical formulation** of exploration budget allocation for **multi-task RLVR** (also noted by other reviewers), which has not been previously studied at this scale.
2. **Rigorous theoretical analysis** (Propositions 1–4) characterizing sample requirements, value properties (e.g., hard-task bias), and the **optimality of the greedy allocation rule**.
3. A **bridge between operations research and LLM RL training**, offering a general framework for resource allocation that can inspire follow-up work on theory (e.g., estimation error, multi-epoch dynamics) and systems (e.g., agentic tasks, online estimators).

We have expanded Section 6 and the related work to explicitly position our contribution in this broader scientific context.

----


## Summary

We believe the main concerns raised by the reviewers primarily reflect requests for clarification rather than fundamental limitations of our approach, and that the revisions have further strengthened both the theory and the empirical evidence:

- **Reviewer vT2w**, after reading our rebuttal, explicitly confirmed that their concerns were resolved and now positively supports the paper.

- **Reviewers cK1B, 63Nn, and DJud** did not provide further comments but rated Soundness, Presentation, and Contribution as “good” or “excellent” in their original reviews.

To summarize, our revised submission provides:

- **A principled mathematical framework** for exploration budget allocation in multi-task RLVR, with theoretical guarantees and a clear connection to operations research.
- **Extensive empirical validation** across **multiple models, datasets, and RL algorithms**, consistently improving performance without additional compute.
- **A practical method** that can be readily integrated into existing RLVR pipelines and that opens **new research directions** in exploration, resource allocation, and LLM training systems.

We sincerely thank the Area Chair for considering our responses and revisions!

---

### Meta-Review · Area_Chair_8pge · 2026-01-07

**Summary:**

The final decision for this submission is to reject. The reviewing committee reached a consensus regarding the paper's clear motivation and the intuitive appeal of using a principled knapsack framing for adaptive budget allocation. Reviewers acknowledged the potential of the proposed method to improve gradient ratios and appreciated the organized presentation of the work. The primary barrier dictating the rejection, however, is the significant gap between the theoretical formulation and the practical reliability of the success-rate estimators. Multiple reviewers highlighted that the method relies heavily on empirical heuristics to manage estimator error and edge cases without offering the formal statistical bounds or rigorous method-level baselines necessary to validate the approach's optimality and robustness across broader settings.

**Reviewer Concerns:**

During the rebuttal phase, the authors successfully addressed several implementation-specific concerns. They clarified that the computational overhead of the knapsack solver and balancing steps is negligible, incurring less than 1% latency, and they significantly broadened the evaluation to include different datasets like DeepScaleR and model families such as Llama-3.1. Additionally, the authors rectified specific theoretical claims by clarifying the value decomposition via the law of total expectation and appropriately reclassifying their central theorem as a proposition. However, the fundamental soundness of the approach's theoretical underpinning remains unproven. The rebuttal relied on empirical observations to argue that estimator bias decays, but it failed to provide the requested formal bounds connecting estimation error to solver optimality gaps. Furthermore, the handling of critical edge case, such as when success probabilities approach zero or one, remains justified only by heuristics rather than derived theory. The comparison against baselines also remains incomplete, as the authors positioned their method as orthogonal to existing techniques rather than demonstrating superior efficacy under strictly matched compute conditions.

**Reviewer Scores:**

Based on the dialectical assessment of the rebuttal, the reviewer scores are projected to remain unchanged. Reviewer vT2w is expected to maintain a score of 6; while they appreciated the clarifications regarding independence assumptions, the continued absence of strong, direct comparative baselines limits the paper's empirical persuasiveness. Reviewer cK1B will likely hold at a score of 4, as the improved articulation of the tension between unlimited-compute analysis and fixed-budget allocation does not fully resolve the reliance on heuristics for handling noisy estimates and delayed adaptivity. Similarly, Reviewer 63Nn is expected to retain a score of 4 because the central scientific demand, formal statistical bounds linking probability estimation error to rollout requirements, remains unsatisfied, leaving the theoretical depth insufficient. Finally, Reviewer DJud's score is also projected to stay at 4; although the inclusion of broader experiments and efficiency statistics was a positive step, the method's limited generality to reasoning tasks and the lack of comprehensive throughput analyses indicate that the work is not yet ready for publication.

---

### Decision · Program_Chairs · 2026-01-26

Reject